# Differences in the Sub-seasonal Predictability of Extreme Stratospheric Events

Rachel W.-Y. Wu[1], Zheng Wu[1], and Daniela I.V. Domeisen[1,2]

[1]ETH Zurich, Zurich, Switzerland
[2]University of Lausanne, Lausanne, Switzerland

**Correspondence:** Rachel Wai-Ying Wu (rachel.wu@env.ethz.ch)

**Abstract.** Extreme stratospheric events such as sudden stratospheric warming (SSW) and strong vortex events can have downward impacts on surface weather that can last for several weeks to months. Hence, successful predictions of these stratospheric events can be beneficial for extended range weather prediction. However, the predictability of extreme stratospheric events is most often limited to around 2 weeks or less. The predictability strongly differs within events of the same type, and also between event types. The reasons for the observed differences in the predictability, however, are not resolved. We extend the analysis of the predictability of stratospheric extreme events to include wind deceleration and acceleration events, with SSW and strong vortex events as subsets, to conduct a systematic comparison of sub-seasonal predictability between events in the European Centre for Medium-Range Weather Forecasts (ECMWF) prediction system. Events of stronger magnitude are found to be less predictable than weaker events for both wind deceleration and acceleration events, with both types of events showing a close to linear dependence of predictability on event magnitude. There are however deviations from this linear behaviour for very strong magnitude events. The difficulties of the prediction system in predicting extremely strong anomalies can be traced to a poor predictability of extreme wave activity fluxes in the lower stratosphere, which impacts the prediction of deceleration events, and interestingly, also acceleration events. Our study suggests that improvements in the understanding of the wave amplification that is associated with extremely strong wave activity fluxes and accurately representing these processes in the model are expected to enhance the predictability of stratospheric extreme events and, by extension, their impacts on surface weather and climate.

## 1 Introduction

The stratospheric polar vortex (SPV) is a band of strong westerly winds over the polar region at the height of around 20-50km during winter. These circumpolar winds result from a strong temperature gradient in the stratosphere between the polar and subtropical regions during winter due to reduced solar heating over the polar regions. As westerly flow in the stratosphere favours upward wave propagation (Charney and Drazin, 1961), planetary-scale waves formed at the troposphere can propagate upwards into the stratosphere (e.g. Polvani and Waugh, 2004; Sjoberg and Birner, 2012). Depending on the wave activity and the state of the vortex, the SPV can undergo periods of weakening or strengthening, thus largely varying in strength during the wintertime.

The weakening and strengthening of the SPV can be understood in the framework of wave-mean flow interaction (Matsuno, 1970; Holton and Mass, 1976). Before vortex weakening events, anomalously strong wave activity is observed in the lower stratosphere (Polvani and Waugh, 2004; Hinssen and Ambaum, 2010). The waves can precondition the vortex via wave breaking (Limpasuvan et al., 2004; Albers and Birner, 2014), shaping the vortex structure to be more favourable for upward wave propagation. A preconditioned vortex is associated with a region of large and positive refractive index (Matsuno, 1970; Simpson et al., 2009; Karoly and Hoskins, 1982). As the refractive index for stationary planetary waves is proportional to the meridional potential vorticity (PV) gradient, the meridional PV gradient can be used as a proxy for waveguidability (Albers and Birner, 2014; Jucker and Reichler, 2018). On the other hand, when wave activity is weak and the SPV is relatively undisturbed, the vortex strengthens on radiative timescales (Limpasuvan et al., 2005; Hitchcock and Shepherd, 2013). Holton and Mass (1976) demonstrated using a simple mechanistic model that when the wave forcing is below a critical level, the vortex accelerates and approaches a state close to radiative equilibrium.

There exist various definitions to characterise the weak and strong states of the SPV. The most commonly studied events are major sudden stratospheric warmings (SSWs, Baldwin et al. (2021)), characterising the abrupt weakening of the SPV. SSW events are commonly defined by the reversal of the SPV mean flow from westerly to easterly (Charlton and Polvani, 2007; Butler et al., 2017; Palmeiro et al., 2015). In some studies, where the primary focus is on the abrupt dynamical nature of SSW events, a definition based on wind change is used (Birner and Albers, 2017; de la Cámara et al., 2019). On the contrary, events where the SPV becomes anomalously strong, with the mean flow accelerating to anomalously strong westerly values beyond a certain threshold, are characterised as strong vortex events (Tripathi et al., 2015). Due to the rapid nature of wave forcing, vortex weakening can be abrupt, whereas vortex strengthening tends to be more gradual (Limpasuvan et al., 2005). The more rapid nature and stronger magnitude of vortex weakening than strengthening can be observed by comparing the magnitude of the identified vortex weakening and strengthening events in studies for SPV variability (e.g. Baldwin and Dunkerton, 2001; Limpasuvan et al., 2005). The asymmetry is also observed in the wave activity preceding the events (Polvani and Waugh, 2004) due to the strong relationship between wave forcing and mean flow.

Weak and strong states of the SPV can have a downward impact on surface weather that can last for a few weeks to a few months (Baldwin and Dunkerton, 2001). This downward influence can potentially be used to extend the predictability limit of surface weather from stratospheric origins (Domeisen et al., 2020a). In the stratosphere itself, the deterministic predictability limit of SSW events is about 10 days (Domeisen et al., 2020b; Taguchi, 2020), and it is found that the predictability of SSWs differs strongly between events (e.g. Karpechko, 2018). The source of predictability of SSW events is attributed in some studies to the predictability of wave activity (Stan and Straus, 2009; Karpechko et al., 2018; Portal et al., 2022) and tropospheric blocking (e.g. Tripathi et al., 2016), as blocking events often precede SSW events (e.g. Martius et al., 2009). It is found in ensemble forecasting systems that when the forecasts are initialised under strong blocking conditions, ensemble members of the forecasts can undergo bifurcation and lead to large uncertainties (Karpechko, 2018; Lee et al., 2019). However, even when successfully predicting a preceding blocking event, a model may still fail to predict a SSW (Tripathi et al., 2016), suggesting that other factors, e.g. the background state of the stratosphere, might be important for successful predictions of SSWs.

Extreme stratospheric events, e.g. SSW and strong vortex events, are often the main focus of stratospheric predictability studies (e.g. Domeisen et al., 2020b; Taguchi, 2014, 2020). Strong vortex events are shown to be more predictable than SSW events (Domeisen et al., 2020b). To our knowledge, the reason for the observed differences in predictability between event types is, however, not resolved in existing literature, and is often attributed to the different mechanisms driving these events. The sample size of SSW and strong vortex events in sub-seasonal prediction systems tends to be too small to systematically assess their differences in predictability. Thus, in this study, we expand the analysis of the predictability of extreme stratospheric events to wind deceleration and acceleration events. As SSW events and strong vortex events are periods of strong zonal wind deceleration and acceleration, respectively, a better understanding of the predictability of wind deceleration and acceleration events will also contribute to the understanding of the predictability of SSW and strong vortex events. We aim to address the following questions: 1. If we expand the event definitions to wind deceleration and acceleration events, do we also see a difference in predictability between wind deceleration and acceleration events, as for SSW and strong vortex events? 2. If so, what contributes to the difference in predictability between events? For example, is predictability related to event magnitude or event mechanisms? 3. What are the dynamical precursors for the predictability of the events? Do those precursors set the predictability limit of the events?

The paper is structured as follows: Section 2 discusses the data and methods adopted in this study. Section 3.1 illustrates the predictability differences between wind acceleration and deceleration events, Section 3.2 discusses the predictability dependence of events on event magnitude, and Section 3.3 explores the predictability dependence on event mechanisms. Finally, we discuss our results in Section 4.

## 2 Data and methods

### 2.1 Datasets

The hindcasts (retrospective forecasts) of the European Centre for Medium-Range Weather Forecasts (ECMWF) model from the subseasonal-to-seasonal (S2S) prediction database (Vitart et al., 2017) are used to evaluate the predictability of stratospheric events in Northern Hemisphere (NH) winter, from November to March (NDJFM), in the period of 1998/99-2017/18, which is the full available hindcast period for the model versions used in this study. The hindcasts are initialised twice a week (every Monday and Thursday) for a 20 year period alongside the real-time operational forecasts. The hindcasts consist of 11 ensemble members.

The model versions CY43R3 and CY45R1, corresponding to hindcasts with model version dates of 2017-07-13 to 2019-06-10, are used. Similar model configurations are used in both model versions, and they both use the ECMWF ERA-Interim reanalysis (Dee et al., 2011) for initialisation. The different model versions lead to qualitatively similar results in terms of prediction skill in their hindcasts (not shown) and are thus both used for the analysis presented here. The hindcasts are verified against the ERA-Interim reanalysis.

We evaluate the skill of the hindcasts at various lead times. Lead time is referred to as the time between the event onset date and the hindcast initialisation date. For example, a lead time of -5 indicates a hindcast initialised 5 days before the event onset.

Hindcasts are divided into 6 lead time groups (LTGs) according to their initialisation dates, each of which represents a 5-day lead time window. For example, LTG-30 refers to hindcasts with initialisation dates of 30 to 26 days before the event, while LTG-5 refers to hindcasts from 5 days to 1 day before the onset date.

## 2.2 Definition of stratospheric events

From the daily mean of the zonal mean zonal wind at $60°$ N and 10 hPa ($\overline{u}$) from NDJFM 1998/99-2017/18 of ERA-Interim, we identify zonal wind acceleration and deceleration events. Both acceleration and deceleration events are defined as 10-day events and are identified using a 10-day moving window. Another event can only be identified 20 days after the start of an event to prevent identifying the same event. If a stronger deceleration is observed within 20 days of the last identified event, the period with stronger wind deceleration is selected instead, replacing the weaker event. The start date of the event is defined as day 0 of the event, i.e. the day when acceleration or deceleration starts in the 10-day window. The magnitude of the identified events is defined as the wind change over the 10-day event window, i.e. $\Delta\overline{u} = \overline{u}(t = 9) - \overline{u}(t = 0)$, where $t$ indicates the lead time. We also impose a criterion that the ratio of the difference between the maximum and minimum wind speed occuring during the 10-day event window to the identified event magnitude has to be less than 1.2, to filter out high frequency variations. Although different processes are involved in deceleration and acceleration events, the duration of wind deceleration and acceleration is found to be similar (Fig. A1a). The 90th percentiles in the duration distributions for both wind deceleration and acceleration are around 10 days. The event magnitude captured by a 10-day window also shows values comparable to the wind changes in SSW and strong vortex events (Fig. A1b). Therefore, after a systematic comparison of different window widths (not shown) and also for comparability between the event types, we use the same event window width of 10 days to identify both wind deceleration and acceleration events.

To compare the identified acceleration and deceleration events with the extreme stratospheric events, we classify the identified events into weak and strong magnitude events. We choose the 60th percentile of event magnitude as the threshold for strong magnitude events. Events that have an absolute magnitude above the respective 60th percentile of the identified acceleration and deceleration events are classified as strong magnitude events, and those below are classified as weak magnitude events. The 60th percentiles are 16.94 ms$^{-1}$ and -24.55 ms$^{-1}$ for the acceleration and deceleration events, respectively, in the reanalysis. In the ECMWF model, the 60th percentiles of the identified events are 16.77 ms$^{-1}$ and -20.87 ms$^{-1}$ for the acceleration and deceleration events, respectively. The thresholds used here are comparable to the thresholds to define strong deceleration events used in other studies (e.g. Birner and Albers, 2017; de la Cámara et al., 2019). Following Birner and Albers (2017), we compute the standard deviation of deseasonalised daily zonal mean zonal wind and the standard deviation ($\sigma$) is found to be around 1 $ms^{-1}/day$. Our chosen 60th percentile from a 10-day wind change corresponds to daily wind changes of 1.69 ms$^{-1}$ and -2.46 ms$^{-1}$ for strong acceleration and deceleration events, respectively, which corresponds to daily wind changes in the 95th and 99th percentile ($1.69\sigma$ and $2.46\sigma$) in NH Nov-Mar. Thus, the strong magnitude events we define here have magnitudes comparable to SSW and strong vortex events.

For the acceleration and deceleration events identified from reanalysis, we check if they are also associated with extreme stratospheric events, i.e. SSWs, strong vortex events and vortex recovery events. SSW events are defined using the Charlton

**Table 1.** Identified acceleration and deceleration events from reanalysis. The numbers in the brackets specify the number of events in each category.

| Acceleration event | Weak (51) | Strong (34) | Total (85) | Definition |
|---|---|---|---|---|
| Strong vortex | 14 | 11 | 25 | Following Tripathi et al. (2015) and Domeisen et al. (2020b) |
| Vortex recovery | 8 | 11 | 19 | $\overline{u}$ at any time during event window shows negative values |
| Other acceleration events | 29 | 12 | 41 | - |
| Deceleration event | Weak (39) | Strong (26) | Total (65) | Definition |
| SSW | 0 | 10 | 10 | Following Charlton and Polvani (2007) |
| Other deceleration events | 39 | 16 | 55 | - |

and Polvani (2007) wind reversal criterion. The onset date of an SSW event is identified as the first day that the daily mean zonal mean zonal winds at 60° N and 10 hPa are negative. The winds have to be westerly for at least 20 consecutive days before the event and return to westerly for at least 10 days after the event. We classify a deceleration event to be associated with an SSW event if an SSW occurs within the 10-day event window. The identified deceleration events can also be associated with early final warming (FW) events. Early FW events are defined as in Butler and Domeisen (2021) as those that occur at least 2 days before the median climatological FW date, which is Apr 12 over the period 1979-2019 in JRA-55 reanalysis. Since we only identify events up to March, the number of events associated with final warming events is small, and wave forcing still plays a dominant role in the FW wind reversal. Therefore, we keep the events associated with final warmings in the analysis and do not distinguish them from other deceleration events.

A strong vortex event is defined when $\overline{u}$ exceeds a threshold value. Following Tripathi et al. (2015) and Domeisen et al. (2020b), the chosen threshold value at 60°N and 10 hPa is 41.2 m/s, which is the 80th percentile of the zonal mean zonal wind averaged from November to March over the 1980-2012 period in ERA-Interim. We classify an acceleration event to be associated with a strong vortex event if the wind at any time during the event window is above this threshold. If the wind at 60° N, 10 hPa at any time during the acceleration event window shows negative wind values, the event is classified as being associated with a vortex recovery event, which occur after SSW events. Table 1 shows the identified events from the reanalysis and their respective event types.

### 2.3 Skill measures

The following metrics are used to assess the predictability of stratospheric events: Mean error, continuous ranked probability score (CRPS), hit-rate (HR), and ensemble spread. The definitions are stated below.

1. Mean error

The mean error is the average difference between the hindcast ($F$) and the observation ($O$) (here, reanalysis is used instead of observations as the verification dataset). The index $i$ denotes the corresponding ensemble member, and N denotes the ensemble size. For the ECMWF model, N = 11. The perfect score of the mean error is 0.

$$Mean\,Error = \frac{1}{N}\sum_{i=1}^{N}(F_i - O_i) \tag{1}$$

2. Continuous ranked probability score (CRPS)

The CRPS measures the difference between the predicted cumulative distribution function (CDF) ($P_f(x)$) of a variable $x$ and the observed CDF ($P_o(x)$). For ensemble forecasts, the predicted CDF is given by the predictions of all the ensemble members. The perfect score of the CRPS is 0.

As the CRPS is given by the difference between the predicted and observed distribution, if all ensemble members in a hindcast predict an event magnitude of 0 $ms^{-1}$, i.e. close to a climatological state where the wind stays relatively constant during a 10-day window, the CRPS of this hindcast will be equal to the observed event magnitude itself.

$$CRPS = \int_{-\infty}^{\infty}(P_f(x) - P_o(x))^2 dx \tag{2}$$

3. Hit-rate (HR)

The hit-rate (HR) is defined as the fraction of ensemble members that successfully predict an event, given by dividing the number of successful members ($M$) by the total number of ensemble members ($N$). A successful prediction requires that the model predicts an event of the same magnitude category as identified from reanalysis, i.e. a strong or weak magnitude event, on the same date as the event in reanalysis. The perfect score of the HR is 1.

$$HR = M/N \tag{3}$$

4. Ensemble spread

The ensemble spread of the ensemble members in a hindcast is measured as the standard deviation of the ensemble member predictions around the ensemble mean ($\overline{F}$). If the ensemble members show perfect agreement with each other, the ensemble spread is 0.

$$Ensemble\,Spread = \sqrt{\frac{\sum_{i=1}^{N}(F_i - \overline{F})^2}{N}} \tag{4}$$

## 2.4 Dynamical indices and significance tests

As mentioned in the Introduction, we can quantify the preconditioning of the vortex background state, which guides waves towards the vortex, by the refractive index. As the refractive index is proportional to the meridional PV gradient ($\overline{q}_y$) divided by the zonal mean zonal wind, following Jucker and Reichler (2018) and Albers and Birner (2014), we approximate the refractive index using the meridional PV gradient. Using the formulation of Equation (5) in Simpson et al. (2009), we divide the meridional PV gradient in spherical coordinates ($\overline{q}_\phi$) by the radius of Earth ($a$) to obtain an equation of the meridional PV gradient in Cartesian coordinates ($\overline{q}_y$),

$$\overline{q}_y = \frac{\overline{q}_\phi}{a} = \frac{2\Omega cos(\phi)}{a} - \left[ \frac{(\overline{u}cos\phi)_\phi}{a^2 cos\phi} \right]_\phi + \frac{f^2}{R_d} \left( \frac{p\theta}{T} \frac{\overline{u}_p}{\overline{\theta}_p} \right)_p \tag{5}$$

where $\phi$ is the latitude, overline denotes the zonal mean, subscripts denote derivatives. As the term associated with Earth's rotation (first term in the equation) is small in extratropical and polar latitudes, and as the third term in the equation correlates well with the second term in the region we consider, i.e. over 55-75° N at 10 hPa (not shown), we use the second term in Equation (5), $- \left[ \frac{(\overline{u}cos\phi)_\phi}{a^2 cos\phi} \right]_\phi$ , as a proxy for waveguidability, hereafter referred to as $\overline{u}_{yy}$. Other than being a reasonable indicator for the refractive index, $\overline{u}_{yy}$ is a measure of the sharpness of the edge of the stratospheric polar vortex, thus also a measure of the strength of the vortex state. Similar to Jucker and Reichler (2018), who used a polar cap averaged meridional PV gradient, we take a latitudinal average of $\overline{u}_{yy}$ over 55-75° N at 10 hPa. As a measure of upward wave activity in the lower stratosphere, following Polvani and Waugh (2004), we use the latitudinal average of meridional eddy heat fluxes ($\overline{v'T'}$) over 45-75° N at 100 hPa, where $v$ is the meridional wind, $T$ is the temperature, and prime ($'$) denotes the departure from the zonal mean. It is, however, important to be aware that the indices $\overline{u}_{yy}$ and $\overline{v'T'}$ might not be independent. To address the interdependency between the indices, we compare the correlations between 10-day averaged $\overline{u}_{yy}$ and 10-day integrated sum of $\overline{v'T'}$ at different time lags. We find the lowest correlation ($r = 0.3$) between $\overline{u}_{yy}$ averaged over days -10 to -1 with respect to the start date of the stratospheric events and $\overline{v'T'}$ integrated over days 0 to 9 during the events (not shown). Thus, we choose to use the time lags mentioned above to examine the predictability of $\overline{u}_{yy}$ and $\overline{v'T'}$ in the following analyses.

We use a one-sample t-test to assess the significance for the mean of a distribution. When comparing the significant difference between two distributions, we use a Kolmogorov-Smirnov test (KS test). For both tests, we use a confidence level of 95%. In our analyses, we use linear regression lines as a reference to compare the relationships between event magnitude and precursors, and between their predictability in the model. It is to be noted that we do not intend to imply that the relationships or the dynamics involved are linear.

## 3 Results

### 3.1 Predictability of stratospheric events in Northern Hemispheric winter

To illustrate the predictability differences between stratospheric events, we compare the skill of the model in predicting different event types as a function of lead time. The magnitude of the events identified in reanalysis ($\Delta \overline{u}$), measured by the wind difference between day 9 and day 0, predicted by the model hindcasts is compared against the same value in reanalysis for all lead time groups (Fig. 1). The left panel in Fig. 1 shows the errors in event magnitude for the deceleration and SSW events (as a subset of deceleration events). The right panel, which has a flipped y-axis, shows the errors in event magnitude for the acceleration events and strong vortex events (as a subset of acceleration events). Values above the zero line indicate an underestimation of the magnitude of both deceleration and acceleration events, while values below zero indicate an overestimation. The box plots in Fig. 1 of most LTGs lie above zero, indicating an underestimation of event magnitude for both acceleration and deceleration events, including strong vortex events and SSWs. The underestimation of the event magnitude reduces towards smaller LTGs. At LTG-5, the model overestimates around 25% of deceleration and 5% of acceleration events, respectively, shown by the bottom of the box and whisker crossing the zero line. The underestimation of deceleration event magnitude is also seen in Karpechko (2018), where the model shows an initial weakening of the vortex but underestimates the event magnitude.

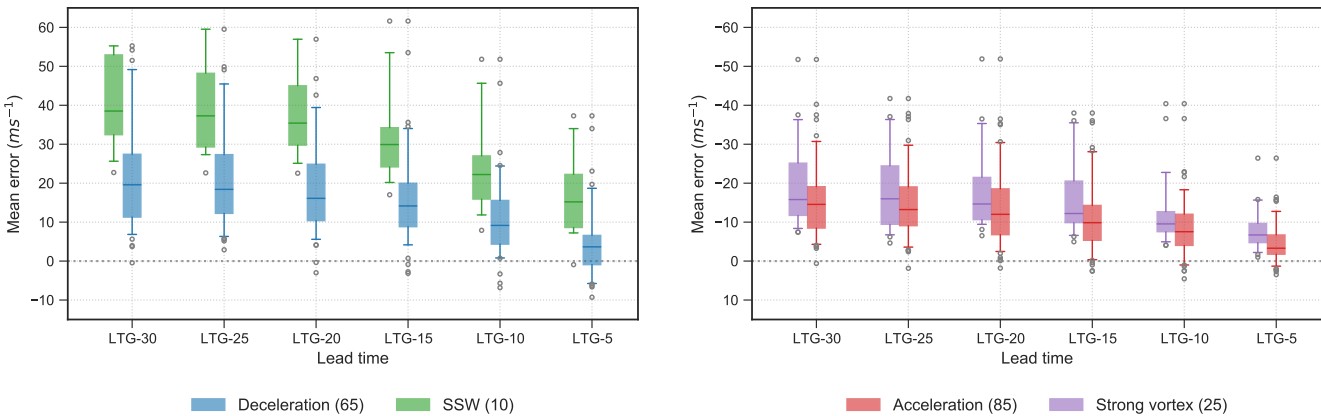

**Figure 1.** Mean error in the magnitude ($\Delta \overline{u}$) of (left) deceleration events (blue), SSW events (green), and (right) acceleration events (red) and strong vortex events (purple), for all LTGs. The y-axis for acceleration events (right panel) is flipped for a more convenient comparison to deceleration events. The box extends from the 25th to the 75th percentiles of the mean error of the events, with a horizontal line at the median. The whiskers extend from the 5th to the 95th percentiles. Outliers are plotted as grey open circles. The numbers in brackets correspond to the number of events in total for each event type in reanalysis.

Previous studies that assessed the predictability of events using event onset dates have found that SSW events are less predictable than strong vortex events (e.g., Domeisen et al., 2020b). This result is confirmed in Fig. 1: The mean errors for SSW events are larger than for strong vortex events, showing that SSW events are less predictable. Extending the analysis to wind

deceleration and acceleration events, we also find that deceleration events are associated with larger errors than acceleration events at all lead times.

## 3.2 Predictability dependence on event magnitude

To better understand the nature of the stratospheric events, we plot the distribution of the events identified from reanalysis (transparent bars in Fig. 2, which are the same in all panels). The events identified from reanalysis show an asymmetry in event magnitude, that is, deceleration events are associated with stronger magnitude than acceleration events. The median magnitude of the wind changes for deceleration and acceleration events in reanalysis is -21.25 ms$^{-1}$ and 15.32 ms$^{-1}$, respectively, and -37.22 ms$^{-1}$ and 15.06 ms$^{-1}$, respectively, for SSW events and strong vortex events. All SSW events belong to the strong deceleration events category, whereas the magnitudes of the strong vortex events are spread more evenly across the weak and strong acceleration event categories (Table 1). The stronger magnitude of deceleration events as compared to acceleration events is consistent with Limpasuvan et al. (2005), i.e. that the daily zonal mean zonal wind anomalies observed for vortex weakening events are stronger than for vortex strengthening events.

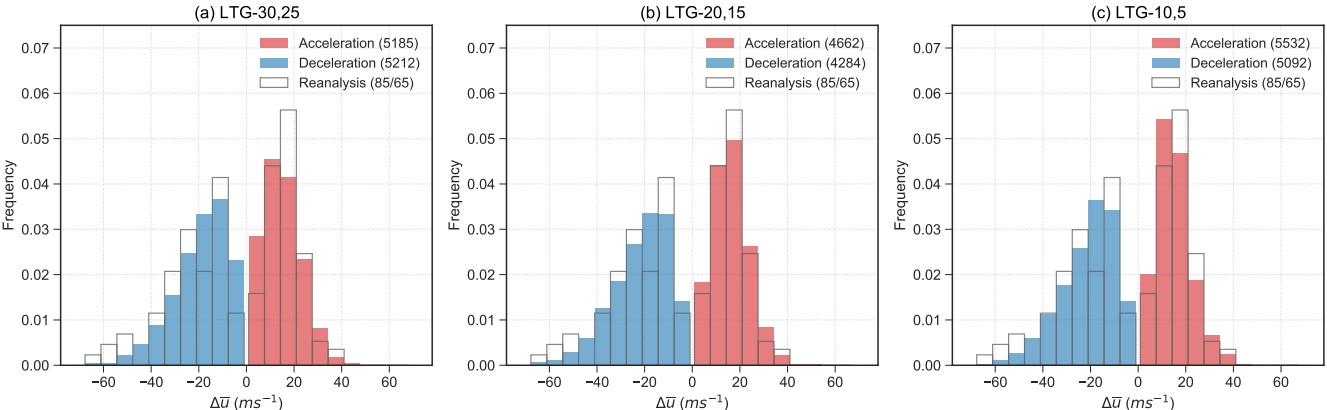

**Figure 2.** Distributions of the wind change ($\Delta \overline{u}$) of the acceleration and deceleration events identified from reanalysis (transparent bars with grey outline) and the acceleration (red) and deceleration (blue) events identified from the ensemble members in the hindcasts that are initialised in NH winter at (a) LTG-30,25, (b) LTG-20,15 and (c) LTG-10,5. Numbers in parentheses indicate the number of identified events at each lead time. The reanalysis distributions displayed in all panels are identical and the numbers in brackets refer to the number of acceleration / deceleration events. The histograms are normalised.

As deceleration events have a stronger magnitude than acceleration events and as the identified events span a wide range of magnitudes, as a first step, we test if the differences in predictability between events arise from different event magnitudes. We plot the CRPS of the model in predicting the event magnitude against the observed event magnitude at different lead times (Fig. 3). A 1:1 grey diagonal line is added to each panel as a guide to compare the skill of hindcasts to the skill of a climatological prediction (see Section 2.3). Points above the diagonal line show a poorer skill than a climatological prediction,

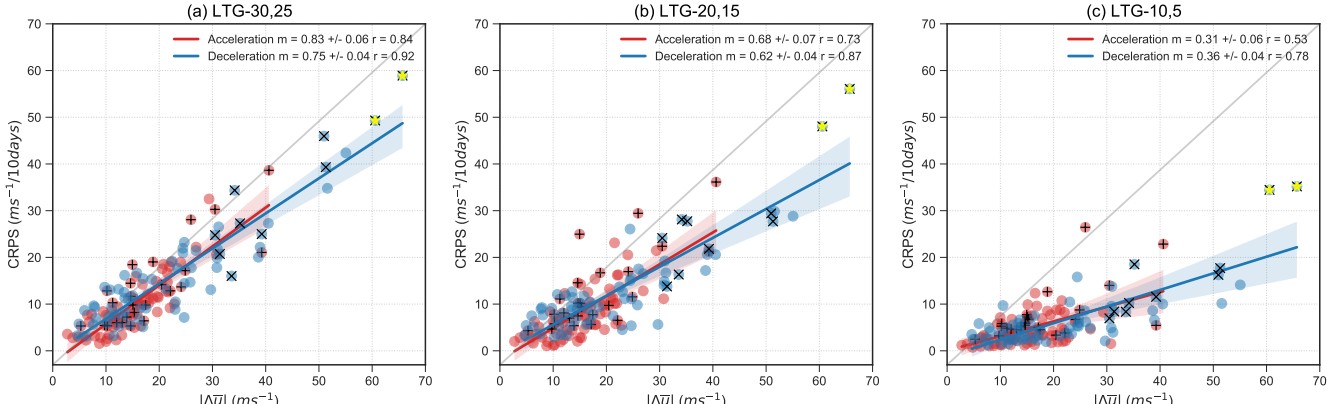

**Figure 3.** CRPS of event magnitude ($\Delta\overline{u}$) for the identified wind acceleration (red) and deceleration (blue) events plotted against their absolute event magnitude ($|\Delta\overline{u}|$) from reanalysis for different LTGs. The absolute value of event magnitude ($|\Delta\overline{u}|$) is used for a better comparison between acceleration and deceleration events. Linear regression lines are fitted to each of the LTGs, $m$ indicates the slope, including the standard error of the fit. Pearson correlation coefficients (r) are indicated in the legend for acceleration and deceleration events, respectively, and r is statistically significant at 95% for all panels. The shaded region shows the 95% confidence interval of the linear fit. Pluses ('+') indicate events that correspond to strong vortex events and crosses ('×') correspond to SSW events. Yellow stars ('*') denote the 2009 and 2018 split SSW events.

and the points below show a skill that is improved with respect to climatology. The closer the points are to the x-axis, i.e. the line of CRPS = 0, the more skilful the hindcasts.

For long lead times of around 30 days, the fitted lines lie just below the diagonal line (Fig. 3a), which suggests that the hindcasts exhibit a predictability that is just slightly better than climatological forecasts at these lead times. The fitted slopes then approach the x-axis with decreasing lead time (going from panels (a) to (c)), indicating that, as expected, the model 235 gains more information from initial conditions and the prediction is improved beyond climatological values. The predictability behaviour of both acceleration and deceleration events can roughly be approximated by a linear fit, indicating that the stronger the event magnitude, the less predictable the event. The linear fits corresponding to the deceleration and acceleration events overlap within the 95% confidence interval (blue and red shading, respectively) at all lead times, suggesting that the acceleration and deceleration events show the same predictability behaviour.

At short lead times, most of the points lie close to zero CRPS. Some events, for instance, the two extreme SSW events with magnitudes of over 60 ms$^{-1}$ (marked by yellow stars in Fig. 3), retain a large CRPS and deviate from the linear fit in the direction of the diagonal line. The fact that the CRPS remains larger for the two events at LTG-5 suggests that the model might not be capturing the precursors or that it might not accurately represent the mechanisms required to predict these events.

To better illustrate the predictability dependence on event magnitude, we composite the strong and weak magnitude events, 245 i.e. events with magnitudes above and below the 60th percentile, respectively, and compare their averaged skill at different lead times (Fig. 4). Overall, as expected from the model capturing more of the required precursors to predict the events, both

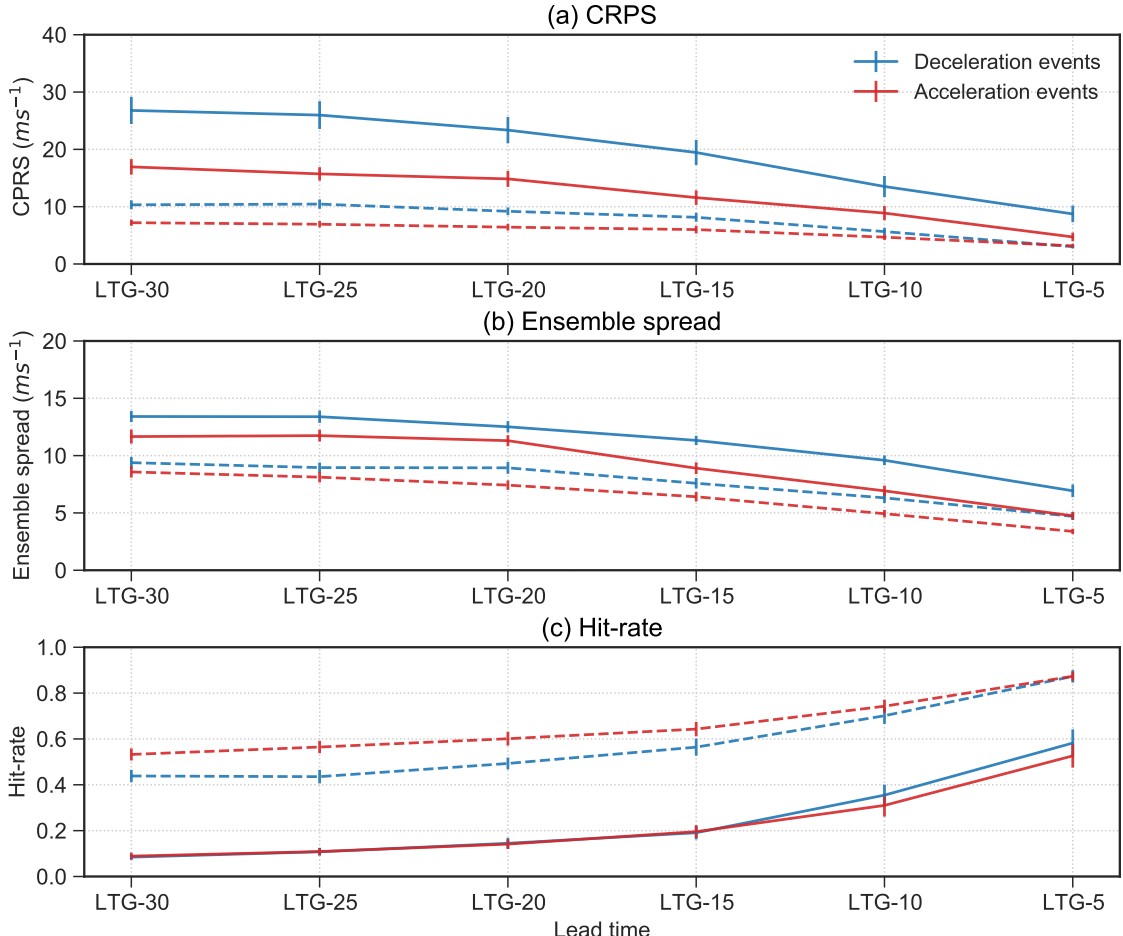

**Figure 4.** (a) CRPS, (b) ensemble spread and (c) hit-rate for acceleration events (red) and deceleration events (blue) computed by validating the hindcasts against the reanalysis. Solid lines indicate the mean of the strong magnitude events at different LTGs and the dotted lines indicate weak magnitude events. The vertical bars indicate the standard errors for each LTG.

acceleration and deceleration events show an increase in hit-rate, and a decrease in ensemble spread and CRPS with decreasing lead time. Strong magnitude events exhibit poorer skill than weak magnitude events, associated with a lower hit-rate, and a larger ensemble spread and CRPS. As large ensemble spread can be observed in ensemble forecasting systems when the forecast is initialised under e.g. strong blocking conditions (Lee et al., 2019; Karpechko, 2018), this might indicate that strong magnitude events are associated with strong precursors or forcings that are not as well captured by the model as those for weak magnitude events. We will discuss the predictability dependence on event mechanism in Section 3.3.

### 3.3 Predictability dependence on event mechanism

In the last section, we showed that event magnitude strongly determines the predictability, with strong events being less predictable, which can be described mostly by a linear behaviour. Some events, however, deviate from this behaviour, which might be connected the mechanism of the events. In this section, we investigate whether the background state of the SPV and the drivers to the events can have an influence on the predictability of events. We start this section by linking the predictability of the events to the related mechanisms through both the influence of the background state of the stratosphere and the drivers in terms of the upward wave flux for both the reanalysis and the prediction system.

### 3.3.1 Vortex background state in reanalysis

Before strong deceleration events, $\overline{u}_{yy}$ is significantly stronger than climatology (Fig. 5a), confirming the existing literature that preconditioning of the vortex via sharpening of the vortex edge is often observed before weak vortex events (e.g. Limpasuvan et al., 2004; Jucker and Reichler, 2018). During strong deceleration events (days 0 to 9), $\overline{u}_{yy}$ reduces to a negative value that is significantly weaker than climatology. The vortex recovers at the end of the strong deceleration event and the mean value of $\overline{u}_{yy}$ returns to positive values, but is still significantly weaker than climatology up to 40 days after the event onset. For weak deceleration events, the values of $\overline{u}_{yy}$ before and after the events are close to climatology, and significant signals are only found during the event window (day 0 to 9), suggesting that the preconditioning of the vortex background state before event onset might not be as important for weak deceleration events. For strong acceleration events, increased $\overline{u}_{yy}$ is found only at around 25 days before the events and a few days later, $\overline{u}_{yy}$ decreases to values significantly lower than climatology (Fig. 5c). During strong acceleration events, $\overline{u}_{yy}$ increases to a value significantly above climatology and drifts back to climatology after the event. For weak acceleration events, a few periods of anomalously weak $\overline{u}_{yy}$ are observed at around day -25, 0, 25.

To further illustrate the relationship between $\overline{u}_{yy}$ and event magnitude ($\Delta\overline{u}$), we plot $\Delta\overline{u}$ against $\overline{u}_{yy}$ averaged over day -10 to -1 for all events (Fig. 6a,d). A significant negative correlation is found between $\overline{u}_{yy}$ and $\Delta\overline{u}$ for deceleration events, that is, the stronger $\overline{u}_{yy}$, the stronger the deceleration. No significant correlation is found for $\overline{u}_{yy}$ averaged over day -10 to -1 against $\Delta\overline{u}$ for acceleration events (Fig. 6d). The averaged $\overline{u}_{yy}$ for acceleration events, however, shows more negative values than for deceleration events. Comparing the distributions of the averaged $\overline{u}_{yy}$ for acceleration and deceleration events, the distributions are found to be significantly different from each other (not shown).

### 3.3.2 Wave activity forcing in reanalysis

In addition to the background state, the forcing by drivers is responsible for extreme stratospheric events. In particular, anomalous wave activity in the lower stratosphere drives the deceleration of the SPV mean flow (Polvani and Waugh, 2004; Hinssen and Ambaum, 2010). The 10-day event window captures well the onset of wind deceleration (Fig. A2a) and the anomalously strong wave activity during the event (Fig. 5b). The wave activity starts to increase from day 0, peaks around day 5 and then decreases to a value that is not significantly different from climatology at the end of the event on day 9. As expected, the wave activity is much stronger during the strong deceleration events than during the weak deceleration events. We find a significant

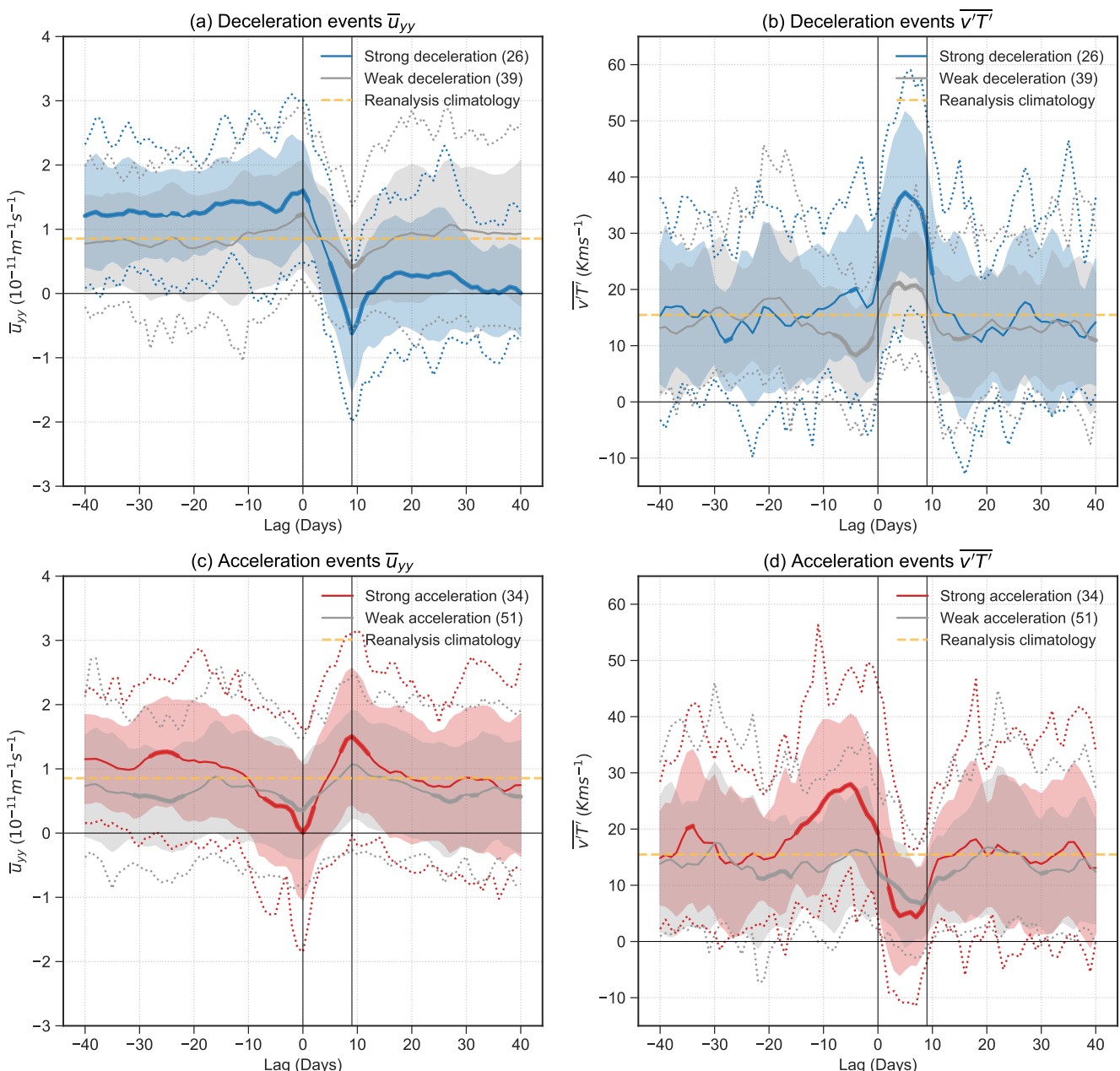

**Figure 5.** Time evolution of daily values of $\overline{u}_{yy}$ (a, c) and $\overline{v'T'}$ (b, d) for the strong deceleration (blue) (a, b) and acceleration (red) (c, d) events in reanalysis. The solid line is the mean value of all events and the bold parts of the line indicate lags where the composites are significantly different from the reanalysis winter climatology (dotted yellow lines) at 95% using a student's t-test. Weak events are composited separately and shown in grey. The dotted lines in the corresponding colours indicate the 5th and 95th percentiles of the composite, the shaded regions indicate the 25th to 75th percentiles. The numbers in the brackets of the legend indicate the number of events in each composite. Lag is relative to the first day of the identified 10-day events.

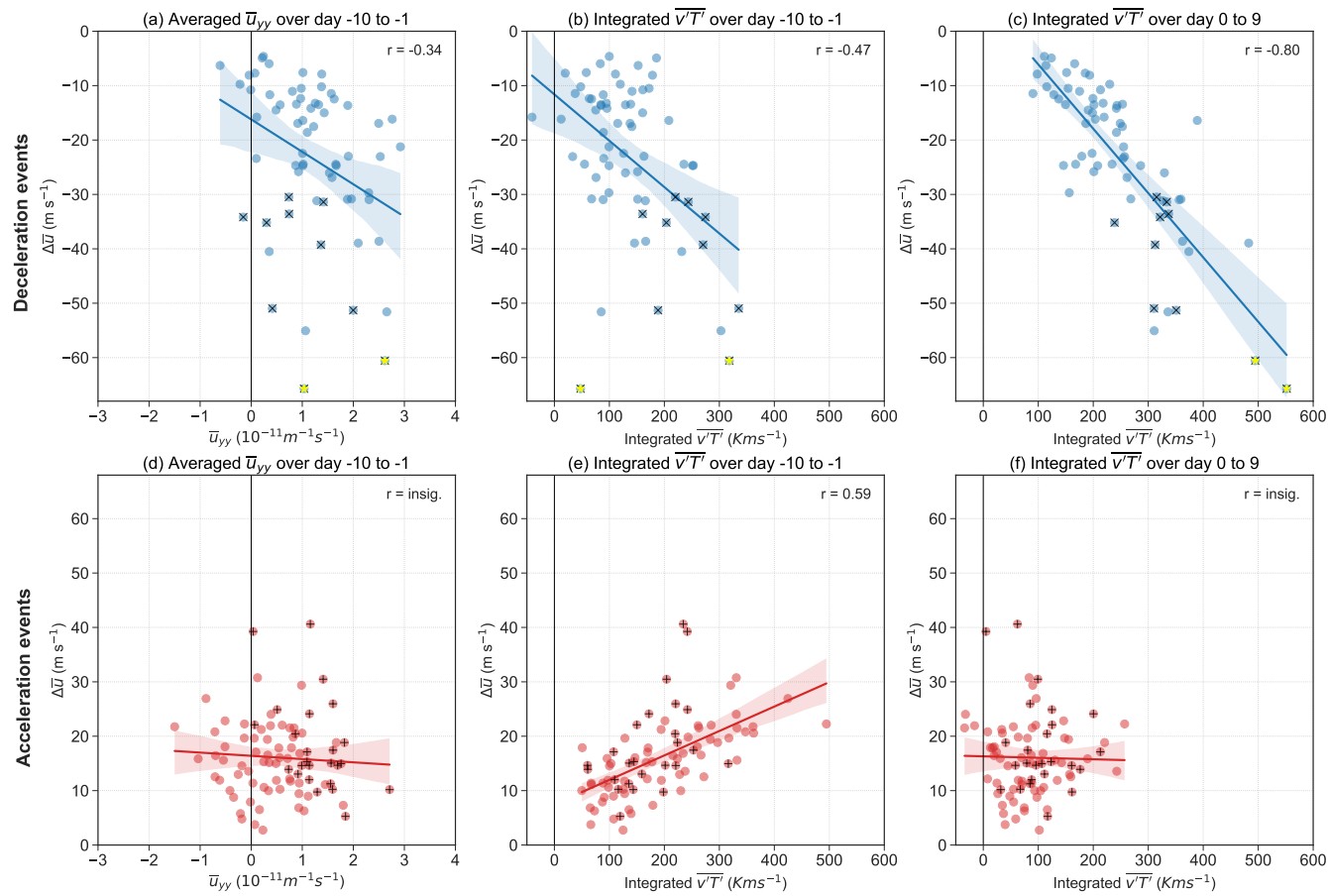

**Figure 6.** Relationship between the magnitude of the deceleration events ($\Delta\overline{u}$) and (a) $\overline{u}_{yy}$ averaged over days -10 to -1, (b) integrated $\overline{v'T'}$ over days -10 to -1 and (c) integrated $\overline{v'T'}$ over days 0 to 9. (d),(e) and (f) same as (a-c) but for the acceleration events. The marker '×' indicates SSW events and '+' indicates strong vortex events. The solid line indicates the fitted linear regression line and shading indicates the 95% confidence interval. Pearson correlation coefficients ($r$) are significant at 95% in all panels. Yellow stars ('*') in (a-c) denote the 2009 and 2018 split SSW events.

negative correlation when correlating the integrated sum of $\overline{v'T'}$ during the event window (day 0 to 9) with the deceleration event magnitude ($\Delta\overline{u}$) (Fig. 6c). The stronger the wave activity during the event window, the more the wind decelerates. For our definition of deceleration events, using a 10-day event window, SSWs occur on average around day 6 of the event window. Therefore, the peak of $\overline{v'T'}$ during the event window is consistent with our understanding that anomalous wave activity precedes SSW events (e.g. Butler et al. (2017)). A wave activity lower than climatology is found around 10 days before (day -10 to -1) the weak magnitude deceleration events but not the strong magnitude events, suggesting the occurrence of weak acceleration events before weak deceleration events. Wind acceleration is indeed observed 10 days before the weak deceleration events, and the magnitude of the acceleration is similar to the magnitude of the subsequent deceleration events (Fig. A2a).

Plotting the integrated $\overline{v'T'}$ for days -10 to -1 against $\Delta\overline{u}$ during the event window, we observe a weak negative correlation (Fig. 6b), which can be explained by low wave activity preceding the weak deceleration events and slightly increased wave activity preceding the strong deceleration events.

Weaker than climatological wave activity $\overline{v'T'}$ is observed during the acceleration event window (Fig. 5d). The wave activity is similar for strong and weak acceleration events but slightly lower for the strong acceleration events. Although strong acceleration events are associated with lower wave activity, there is no significant relationship between the integrated heat flux and event magnitude (Fig. 5f), indicating that wave activity does not drive the acceleration event magnitude, and low wave activity might be more of a threshold criterion for an acceleration event to occur. The passive role of wave activity in wind acceleration is consistent with our understanding that radiative cooling drives the wind acceleration when the wave activity is below a critical level. Interestingly, we observe strong heat flux from around 15 days before the strong acceleration events (Fig. 5d). The same is observed when we exclude vortex recovery events in the composite (not shown). We find a significant positive correlation between the integrated $\overline{v'T'}$ for days -10 to -1 and the wind change over the acceleration event window (day 0 to 9) (Fig. 6e). We find that deceleration events precede about 74% of the strong acceleration events (not shown). The deceleration events that happen before the acceleration events can weaken the vortex, preconditioning the background state of the vortex to be more favourable for the onset of acceleration events, consistent with the weakening of $\overline{u}_{yy}$ before the onset of strong acceleration events from around day -10 (Fig. 5c). The alternation between deceleration and acceleration events is reminiscent of the characteristics of stratospheric vacillations as described in the Holton-Mass model (Holton and Mass, 1976), which shows an oscillation of the mean flow of the vortex after an initial wave forcing.

It is interesting to note that the two strongest strong vortex events, the events with a magnitude of around $40\,\mathrm{ms}^{-1}$, are further away from the linear fit, suggesting that factors other than low wave activity might play a role for these strong magnitude events, for example, strong ozone depletion (e.g., Haase and Matthes, 2019; Lin et al., 2017).

### 3.3.3 Representation of dynamical processes in the model

Deceleration and acceleration events are found to be driven by anomalies in $\overline{u}_{yy}$ and $\overline{v'T'}$ as described in Section 3.3.1 and 3.3.2 using reanalysis. We now assess the ability of the model to represent these anomalies and relationships. We start by treating all ensemble members independently and identify deceleration and acceleration events from each separate member at different lead times.

Overall, the climatology of the model event magnitude is similar to that observed in reanalysis at all lead times (Fig. 2). Using a KS test for the model distributions for acceleration and deceleration events, respectively, against the reanalysis distributions, the model and reanalysis distributions are found to not be significantly different from each other. A similar number of events is identified at all lead times, but slightly fewer at LTG-20,15. At LTG-30,25, the model shows an overall underestimation of event magnitude and produces more events with a magnitude close to zero in the model as compared to reanalysis, which is consistent with the predictions being close to climatology at long lead times (Fig. 3a). The number of events with very weak magnitude decreases when events are identified at shorter lead times. At all lead times, the model underestimates the number of extremely strong deceleration events (shown by the difference between the model and reanalysis in the negative

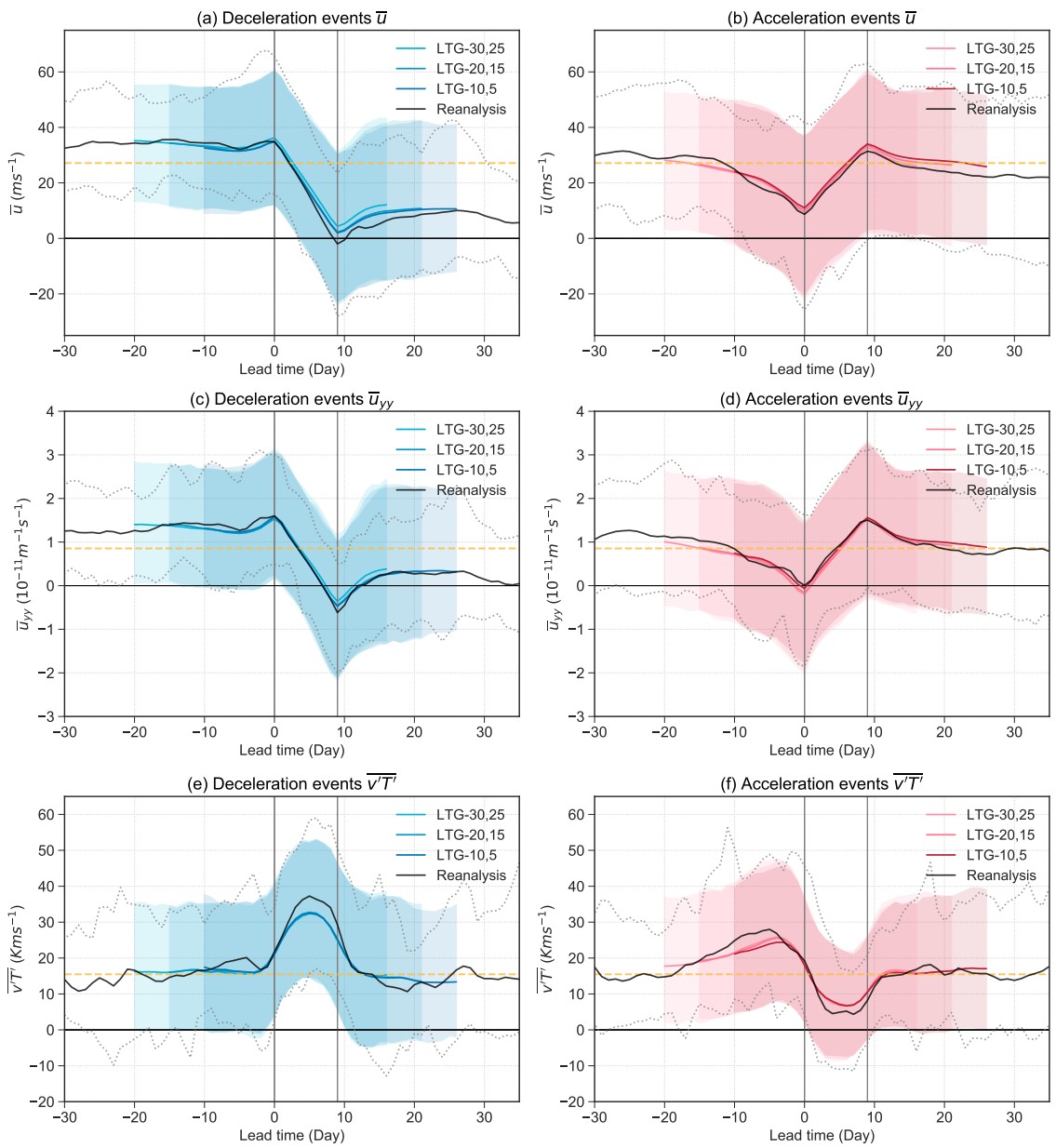

**Figure 7.** Temporal evolution of $\overline{u}$, $\overline{u}_{yy}$, and $\overline{v'T'}$ for (a),(c),(e) the strong deceleration and (b),(d),(f) the strong acceleration events identified in the model at different LTGs. Solid lines indicate the mean of the event composites and shadings indicate the 5th and 95th percentiles of the events in the prediction system. Black solid lines and black dotted lines indicate the mean, 5th and 95th percentiles for the reanalysis, respectively. Yellow dotted lines show the winter climatology in reanalysis. The first and last 5 days in the mean evolution in the model composite are discarded to account for the different start dates within each LTG.

tails of the distributions). The model covers the range of acceleration event magnitude well but underestimates the frequency of acceleration events with moderate magnitude, i.e. around magnitudes of 20 ms$^{-1}$ over the 10-day event window.

To assess the ability of the model in representing the event mechanisms, we composite the identified strong magnitude events from the model and compare the model evolution of the dynamical variables to the observed evolution in reanalysis (Fig. 7). The model shows a time evolution of $\overline{u}$ similar to that from reanalysis. However, as the model underestimates the number of deceleration events with extremely strong magnitude (Fig. 2), the mean evolution of $\overline{u}$ for deceleration events at all lead times in the model stays above zero, while the winds in reanalysis cross the zero wind line (Fig. 7a). The 5th and 95th percentiles of the model events (shadings) shifted towards more positive $\overline{u}$ as compared to reanalysis (dotted lines) around the end of the event window, indicating that the model does not reach values of $\overline{u}$ that are as small as observed in reanalysis. The mean evolution of $\overline{u}$ at LTG-30,-25 (lightest blue) remains above the values for all other LTGs throughout the event window until the end of the forecast.

The vortex background state is well represented in the model at all lead times for both strong deceleration and acceleration events. The model shows near identical mean values and variability comparable to the reanalysis for events identified at all lead times (Fig. 7c,d). For the wave forcing (Fig. 7e,f), the model events do not show the extremely high values of $\overline{v'T'}$ during strong deceleration events, or the extremely low values of $\overline{v'T'}$ during strong acceleration events, at all lead times. The 95th percentile of $\overline{v'T'}$ for the deceleration events in reanalysis is outside of the 95th percentile of the model (colour shadings). Similarly, for the acceleration events, the 5th percentile of the reanalysis composite is outside that of the model. Before acceleration events, a peak of $\overline{v'T'}$ is also observed in the model. However, the wave activity in the model for this peak before acceleration events peaks at a later time and at a lower magnitude.

Given the strong relationship observed between event magnitude and wave activity for deceleration events in reanalysis (Fig. 6c), the observed underestimation of strong $\overline{v'T'}$ for deceleration events in the prediction system might explain the observed underestimation of model deceleration event magnitude in Fig. 7a. As a sensitivity experiment, Fig. 7a and 7e are re-plotted by excluding the events with magnitude above the 90th percentile from the reanalysis composite of strong deceleration events (Fig. A3). It is found that the averaged evolution of $\overline{u}$ and $\overline{v'T'}$ of the model composite then covers almost the full variability of the re-computed reanalysis composite, and the evolution of the model composite is near identical to the reanalysis composite and covers almost the full range of the 5th and 95th percentiles. This suggests that the model has limitations in producing events that have equivalent event magnitudes of above the 90th percentile of the reanalysis deceleration events, likely due to not producing the required strong wave activity. The model also does not show as low $\overline{v'T'}$ during strong acceleration events (Fig. 7f). Although we see the model can produce acceleration events with an evolution similar to reanalysis (Fig. 7b), showing a good variability of acceleration event magnitude in the model, the frequency of acceleration events with moderate event magnitude might still be underestimated (as earlier discussed in Fig. 2). As discussed in Section 3.3.2, $\overline{v'T'}$ might be more a threshold criterion for acceleration events to occur. Specifically, if the wave activity produced in the model is not low enough in some occasions, this can contribute to an underestimation in the number of acceleration events with moderate magnitude, consistent with Figure 2.

We found that overall the model is able to produce events with a range of magnitude similar to reanalysis and has a good representation of event mechanisms. The model, however, has limitations in producing extremely strong anomalies in heat fluxes, thus might be underestimating the number of moderate magnitude acceleration events and the number and magnitude of extremely strong deceleration events. To elucidate the sources of predictability for the events, we now evaluate the magnitude of the anomalies in the precursors, i.e. in $\overline{u}_{yy}$ and $\overline{v'T'}$, captured by the model when predicting the events identified from reanalysis (Fig. 8). The lead time shown in Fig. 8 (and Fig. 9) is with respect to the start date of the events. Since we are taking the day -10 to -1 average of $\overline{u}_{yy}$ and the lead time is defined to be with respect to the start date of the event, day -10 to -1 is out of the time range of hindcasts with lead time of 10 days or less. Thus, we do not have data and to show plots for LTG-10,5 for $\overline{u}_{yy}$.

At LTG-30,25, the anomalies in the precursors captured by the model are weak, indicated by the predicted distributions for both acceleration and deceleration events centering around the climatological values (Fig. 8e,h), which is reflected in the fact that acceleration and deceleration events at these lead times are barely separated (Fig. 8a). This is also consistent with Fig. 3 that the points lie close to the diagonal line at long lead times. Nevertheless, the predicted distributions of $\Delta\overline{u}$ are skewed towards the correct signs of the observed events (e.g. the predicted wind change for deceleration events is skewed towards negative values) (Fig. 8a). The predicted distributions for the precursors of acceleration and deceleration events are significantly different from each other, for all lead times, showing that the magnitude of the precursors captured for acceleration and deceleration events are statistically distinguishable even at long lead times.

For shorter lead times, the predicted distributions for acceleration and deceleration become more clearly distinct as the difference between the predicted deceleration and acceleration distribution increases (indicated by greater distance between the distribution means and higher KS test score at shorter lead times). At LTG-20,15, the model already shows a distribution in $\overline{u}_{yy}$ that is qualitatively similar to reanalysis (Fig. 8f,g), while this is not the case for integrated $\overline{v'T'}$ (Fig. 8i,k) and $\Delta\overline{u}$ (Fig. 8b,d). Even at LTG-10,5, the model shows a clear underestimation of the very strong deceleration events (with $\Delta\overline{u}$ stronger than -40 ms$^{-1}$) and an underestimation in the very high values of integrated $\overline{v'T'}$. The frequency of values with around 400 $mKs^{-1}$ are underestimated and values above 400 $mKs^{-1}$ are scarcely predicted in the model (Fig. 8j,k). For acceleration events the distribution of the wind change even includes negative values in the predicted event magnitude distribution at LTG-10,5 (Fig. 8c), which is not the case in reanalysis (Figure 8d). As a result, the mean of the predicted $\Delta\overline{u}$ distribution is lower than for reanalysis. The model predicts more acceleration events with $\Delta\overline{u}$ close to 0 ms$^{-1}$/ 10 days and some acceleration events with negative $\Delta\overline{u}$. The model also shows a shift to larger values in integrated $\overline{v'T'}$ than that in the reanalysis (Fig. 8j and 8k).

To quantify the contribution of the predictability of the precursors to the predictability of event magnitude at different lead times, we plot the CRPS of the event magnitude against the CRPS of the precursors (Fig. 9). A significant correlation is found between the CRPS of the event magnitude and of the precursors for both acceleration and deceleration events at all lead times. Consistent with Fig. 8, the model captures the anomalies of the precursors more accurately with decreasing lead time. Specifically, as the CRPS in $\overline{u}_{yy}$ and integrated $\overline{v'T'}$ decreases, the CRPS in $\Delta\overline{u}$ also decreases. On the other hand, the CRPS of $\Delta\overline{u}$ shows a stronger correlation with the CRPS of integrated $\overline{v'T'}$ than $\overline{u}_{yy}$, indicating a stronger contribution of the

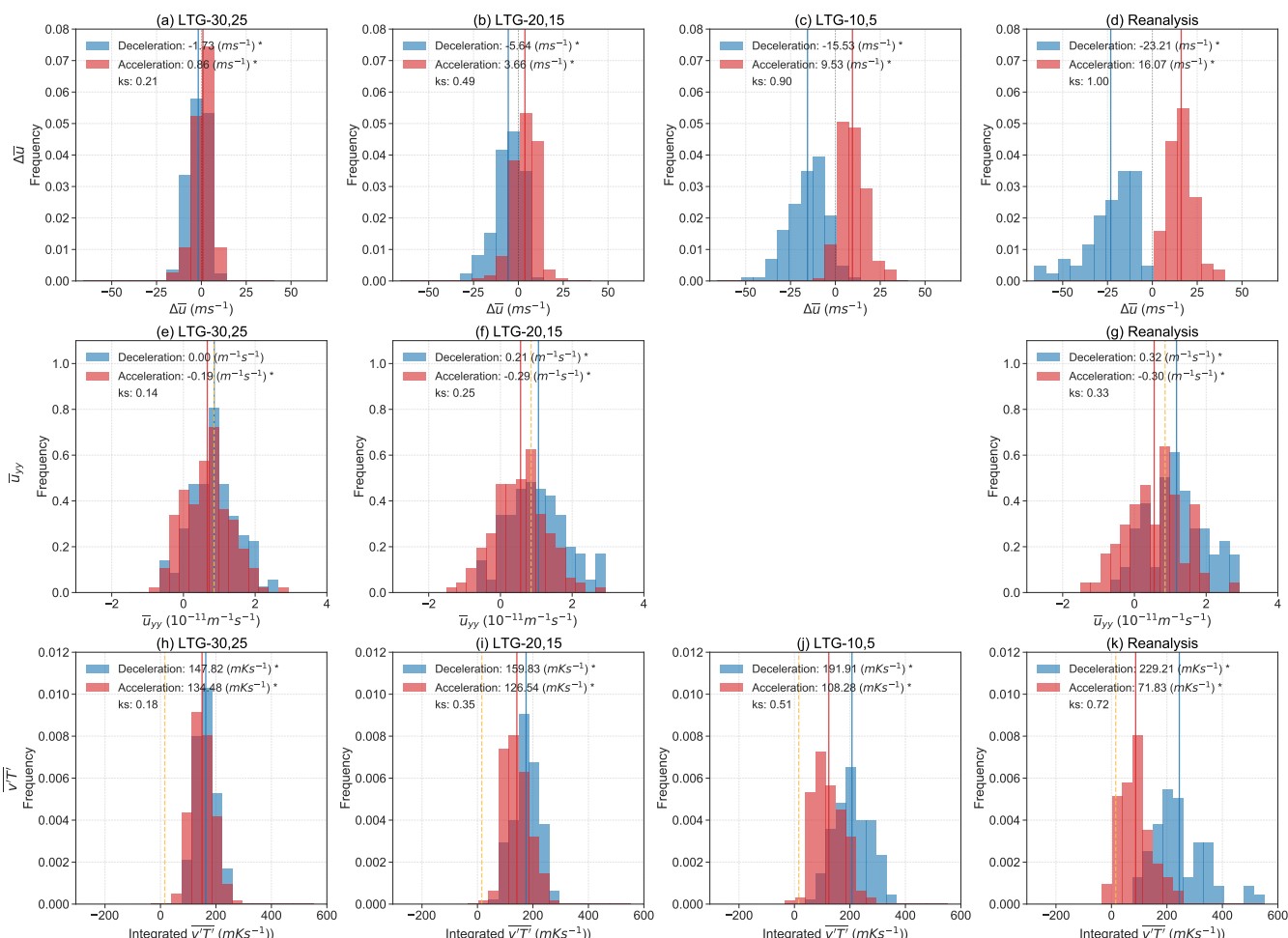

**Figure 8.** Ensemble mean values of $\Delta\overline{u}$ (a-c), $\overline{u}_{yy}$ averaged over days -10 to -1 (e,f) and integrated $\overline{v'T'}$ over days 0 to 9 (h-j) predicted by the model for the events diagnosed in reanalysis at different lead times. The observed distributions from reanalysis are shown in panels (d),(g) and (k). Deceleration events are shown in blue and acceleration events in red. The winter climatological values of $\overline{u}_{yy}$ and integrated $\overline{v'T'}$ from reanalysis are plotted as yellow dotted lines. The differences of the mean of the distributions from 0 (for $\Delta\overline{u}$) or from climatology (for $\overline{u}_{yy}$ and integrated $\overline{v'T'}$) are shown in the legend. * indicates when the distributions are significantly different from 0 in (a) to (d) and when the distributions differ from the reanalysis climatological value in (e) to (l) using a t-test. The histograms are normalised and a KS test is used to test the significant difference between the acceleration and deceleration event distributions. KS statistics ($ks$) are all statistically significant.

predictability of integrated $\overline{v'T'}$ to the predictability of event magnitude, which is consistent with the more direct role of $\overline{v'T'}$ than $\overline{u}_{yy}$ in forcing the events.

The largest CRPS values in $\overline{v'T'}$ are found for the 2009 and 2018 split SSW events (yellow stars in Fig. 9). These two SSW events were associated with very strong wave-2 activity (Ayarzagüena et al., 2011; Domeisen et al., 2018). The large $\overline{v'T'}$ for

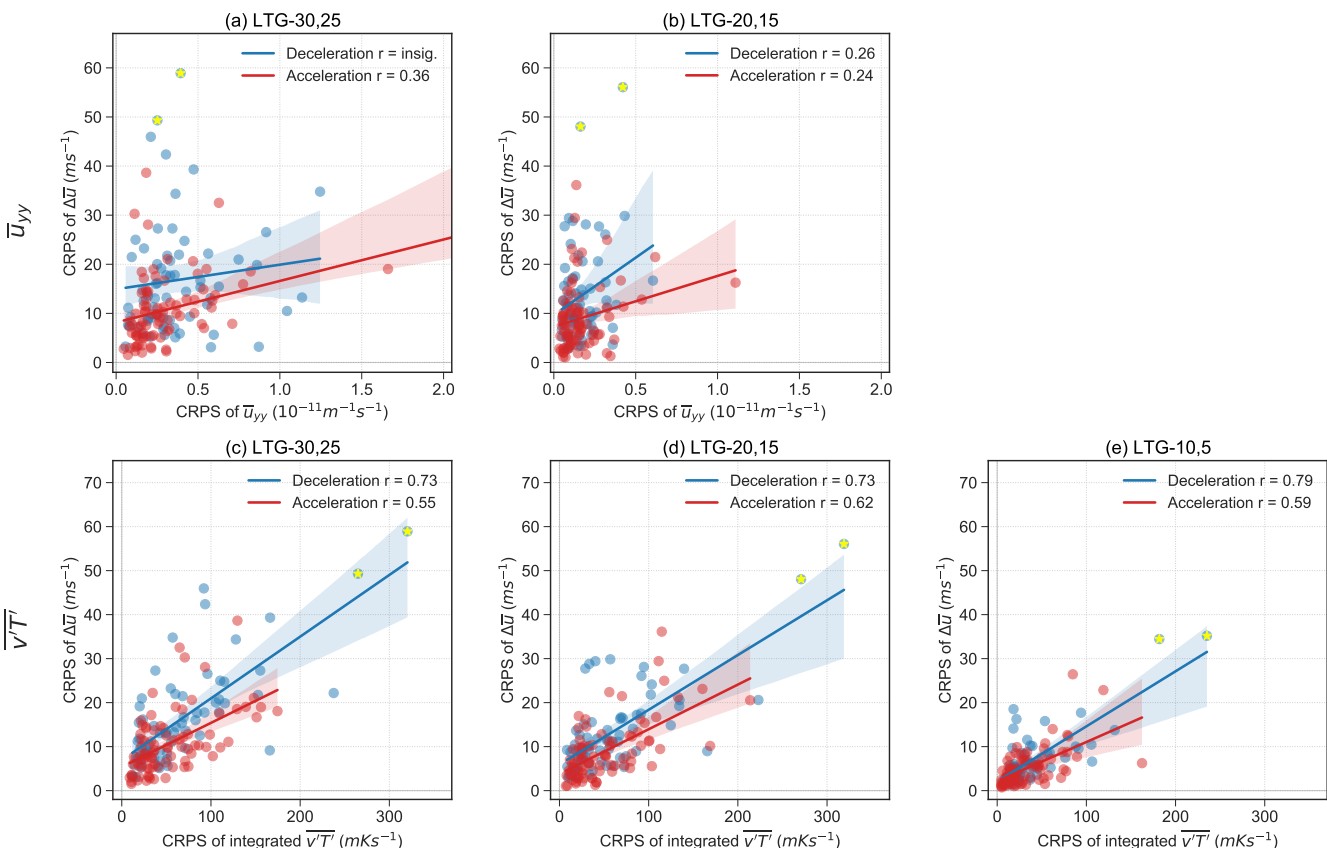

**Figure 9.** Relationship between the CRPS of $\Delta\overline{u}$ with (a-b) the CRPS of $\overline{u}_{yy}$ on day -10 to -1 and with (c-e) the CRPS of integrated $\overline{v'T'}$ over days 0 to 9 for the acceleration (red) and deceleration events (blue) identified from reanalysis. The solid line and the shading correspond to the fitted slope and 95% confidence interval of the fit. The Pearson correlation coefficients ($r$) indicate the correlation in the scatter plots and are statistically significant in all panels at the 95% level. Yellow stars ('*') denote the 2009 and 2018 split SSW events.

these events might be out of the range of $\overline{v'T'}$ that the model can produce, as suggested earlier, or the mechanisms for these events may not be properly represented in the model. We further investigate the regional origin of the $\overline{v'T'}$ errors by dividing up the regions of heat flux origin into Northern Europe, Siberia, North Pacific and North America/ Greenland (Fig. A4). All regions contribute to the errors, with the largest contribution coming from the North Pacific, and smaller contributions from Northern Europe and Siberia (Fig. A5 and A6). Additional analysis is needed to further understand the origin of the $\overline{v'T'}$ errors.

## 4 Conclusions

By expanding the stratospheric event definition to wind deceleration and acceleration events using the tendency of the zonal mean zonal wind at 60° N and 10 hPa, we systematically investigate the predictability of extreme events in the SPV in the

ECMWF S2S hindcasts. We demonstrate that, overall, the ECMWF model represents the variability of the SPV well in terms of event magnitude and the associated dynamical drivers, and it has a good representation of the dynamical processes that are observed in reanalysis. The model, however, shows limitations in producing events with extremely strong deceleration magnitudes. We find that this is associated with the inability of the model to produce extremely strong wave activity in the lower stratosphere.

The large number of identified deceleration and acceleration events allows us to robustly compare the differences in the event mechanisms in both reanalysis and the model, and to understand the differences in the predictability between events. Consistent with our understanding of the mechanisms of wind deceleration and acceleration events in the framework of wave-mean flow interaction, we find that deceleration and acceleration events are associated with the same anomalies but of opposite signs, namely a strengthened waveguide, in terms of the second meridional derivative of the zonal wind ($\overline{u}_{yy}$), and higher wave activity for deceleration events, measured by the 100 hPa eddy heat flux ($\overline{v'T'}$), and vice versa for acceleration events. The predicted distributions of the acceleration and deceleration events become more distinct at shorter lead times and the respective characteristics of the distributions become better represented. For example, the long tails of deceleration events towards strong events become better represented, although the model continues to underestimate these long tails, even at short lead times.

A large part of the predictability differences between events can be explained by the different event magnitudes. When we express the predictability of deceleration and acceleration events in terms of event magnitude, we found that they both show a predictability dependence on event magnitude; that is, events of stronger magnitude are less predictable. We explain the observed predictability dependence from two perspectives: 1) In a statistical sense, strong magnitude events lie within the tails of the climatological distribution and are penalised more heavily than weak magnitude events, and 2) from a dynamical perspective, strong magnitude events are associated with strong anomalies in $\overline{v'T'}$ and $\overline{u}_{yy}$. The strong precursor anomalies are often less predictable in the model and thus can lead to large uncertainties in event magnitude. The same predictability behaviour with respect to event magnitude for deceleration and acceleration events thus suggests that the observed predictability difference between the event types can to a large extent be explained by the difference in event magnitude between the event types, i.e. the fact that wind deceleration events are associated with greater magnitudes than wind acceleration events, and that SSW events are stronger in magnitude than strong vortex events. We also show that the predictability of the $\overline{v'T'}$ and $\overline{u}_{yy}$ can explain most of the predictability of the events, with $\overline{v'T'}$ contributing a larger part of the predictability as compared to $\overline{u}_{yy}$. The predictability limit of these dynamical precursors might, therefore, set the predictability limit of events.

Further work is needed to understand the potential reasons as to why the model has limitations in producing extremely strong wave activity. Events preceded by strong wave activity often coincide with large errors remaining in the prediction even at short lead times. For example, the split SSW events in 2009 and 2018, the events with the two strongest event magnitudes of all deceleration and acceleration events investigated in this study, are preceded by anomalously strong wave-2 wave activity (Harada et al., 2010) and are reported to be more unpredictable than other SSW events (Rao et al., 2018). The large errors associated with certain events even at short lead time suggest that these events might be associated with mechanisms that are different from weaker magnitude events. For example, internal stratospheric dynamics might play a more important role (e.g. Plumb, 1981; Matthewman and Esler, 2011; Birner and Albers, 2017; Domeisen et al., 2018), which might not be well

represented in the model. In fact, the ability of the model to capture the nonlinear dynamics, which are known to be relevant in particular for SSWs with strong magnitude, has not been explored in this study. These nonlinear processes include the complex behaviour of wave breaking, which depending on its exact location and temporal variability can have different effects on the polar vortex, for instance, high frequency wave activity can strengthen the polar vortex rather than weakening it (Harnik, 2009). Furthermore, vortex preconditioning may also be an important factor in determining predictability. For example, certain geometrical configurations of the initial state of the vortex might be more susceptible to vertical wave propagation and weak breaking (Albers and Birner, 2014; Matthewman and Esler, 2011; Esler and Matthewman, 2011). Remote precursors from the tropical stratosphere (Garfinkel et al., 2018; Gray et al., 2020, 2022), the tropical troposphere (Domeisen et al., 2015; Garfinkel and Schwartz, 2017), and the extratropical troposphere (Martius et al., 2009; Karpechko et al., 2018; White et al., 2019; Peings, 2019) can also further enhance predictability on sub-seasonal to seasonal timescales. As such, one might want to investigate how the 2009 and 2018 split SSW events differ from other deceleration events in terms of their preconditioning processes, and to see whether the mechanisms, in particular the nonlinear processes, associated with the two events are well represented in the model. A better representation of the wave amplification mechanisms and extremely strong wave activity in the model can potentially enhance the predictability of stratospheric events, and by extension their impacts on surface weather and climate.

## Appendix A

To choose a suitable event window width for identifying acceleration and deceleration events, we study the variability of the SPV through the tendency in the zonal mean zonal wind at 60° N, 10 hPa in reanalysis. We identify periods that show consecutive days of wind acceleration and deceleration by counting the number of consecutive days that the daily wind change is of the same sign. If the wind changes sign on one day, that day is counted as a new period of wind change. The number of days in the identified wind change period is defined as the duration.

The duration distribution for wind acceleration and deceleration is qualitatively similar to each other, both following an exponential distribution (Fig. A1a). The magnitude is given by the wind change over the duration of a given identified period. The duration and event magnitude shows a close to linear relationship (Fig. A1b).

As an extra analysis, we have separated the reanalysis data of meridional heat flux into contributions from four regions that are selected based on the existing literature and on our own analysis. Specifically, based on the meridional heat flux composite of deceleration events (Fig. A4a), we divided the 45-75°N latitude region equally into four regions as indicated in Figure A4: (a) Northern Europe (40°W - 50°E), (b) Siberia (50°E - 140°E), (c) North Pacific (140°E - 130°W) and (d) North America/ Greenland (130°W - 40°W). The composites with respect to day 0 to 9 of the deceleration events in comparison to the November to March average show anomalously positive heat flux in the three regions, namely, Northern Europe, Siberia and North Pacific, and anomalously negative averaged heat flux in the region North America/ Greenland. The composite for acceleration events shows similar patterns. Thus, we choose to average the heat flux over the same four regions for both deceleration and acceleration events to examine the predictability of the wave activity captured by the model at different lead times.

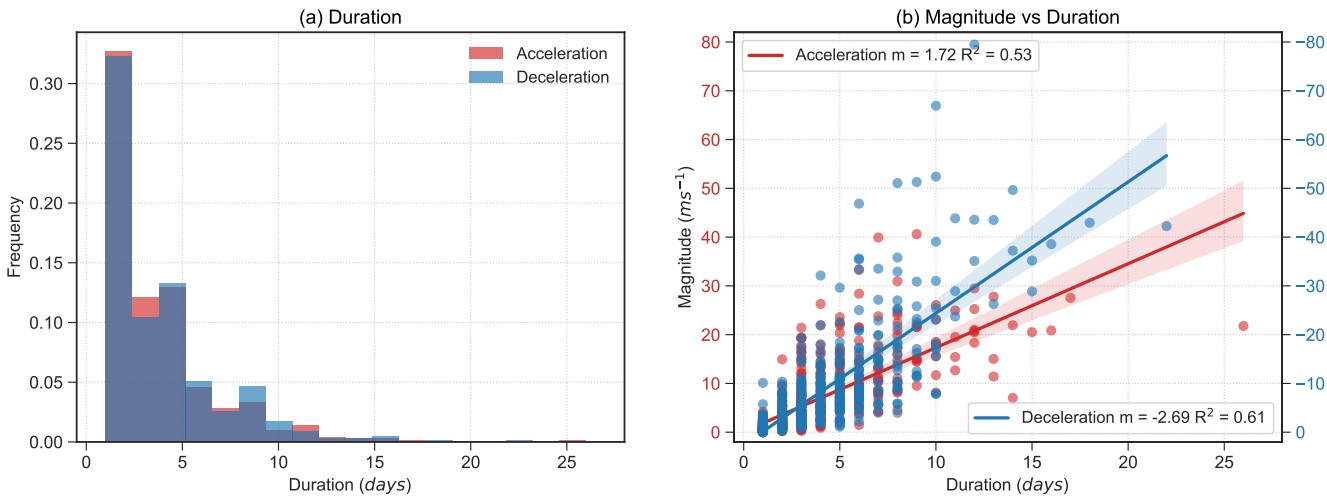

**Figure A1.** Periods of wind acceleration (red) and deceleration (blue) in reanalysis. (a) Duration, (b) the relationship between the magnitude and duration of the wind acceleration and deceleration periods. For acceleration periods (red), refer to the red axis on the left. For deceleration periods (blue), refer to the blue axis on the right. The solid lines mark the linear fit to the scatter plots and the shading marks the 95% confidence interval of the fit. The histograms are normalised.

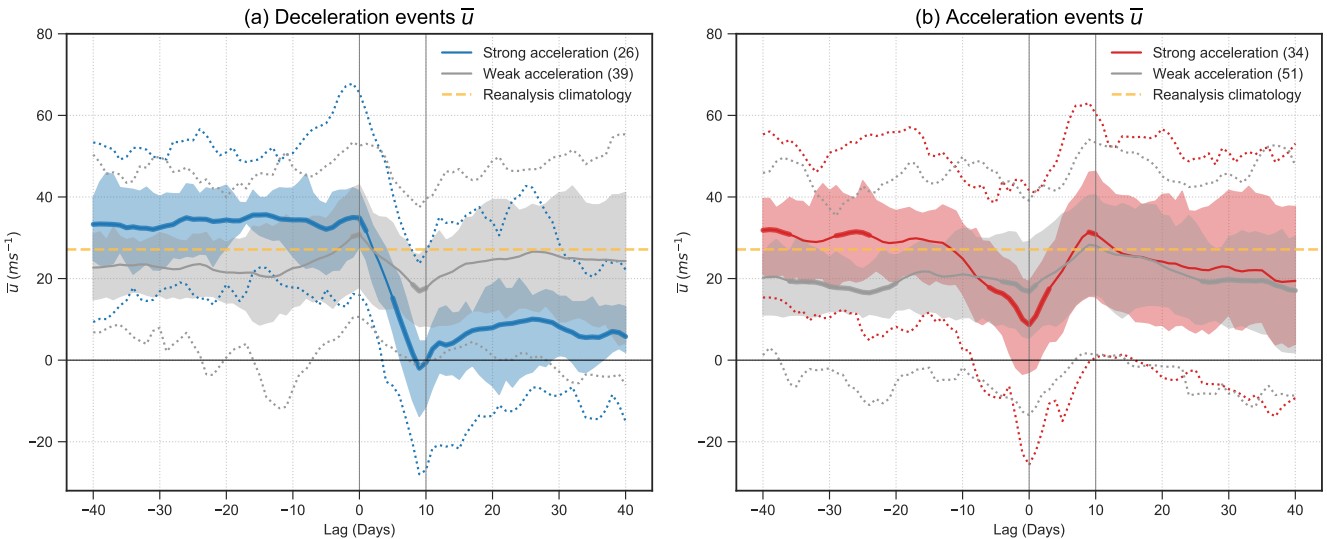

**Figure A2.** Time evolution of daily values of $\overline{u}$ for the strong deceleration (blue) and acceleration (red) events in reanalysis. The solid line shows the mean value and the bold line indicates where the composites are significantly different from the reanalysis climatology using a student's t-test. Weak events are composited separately and shown in grey. The dotted lines in the corresponding colours indicate the 5th and 95th percentile of the composite, the shaded region indicates the 25th to 75th percentiles. The dotted yellow line shows the winter climatology $\overline{u}$ in reanalysis. The number in the brackets of the legend indicates the number of events in the composites. Lag is relative to the first day of the identified 10-day events.

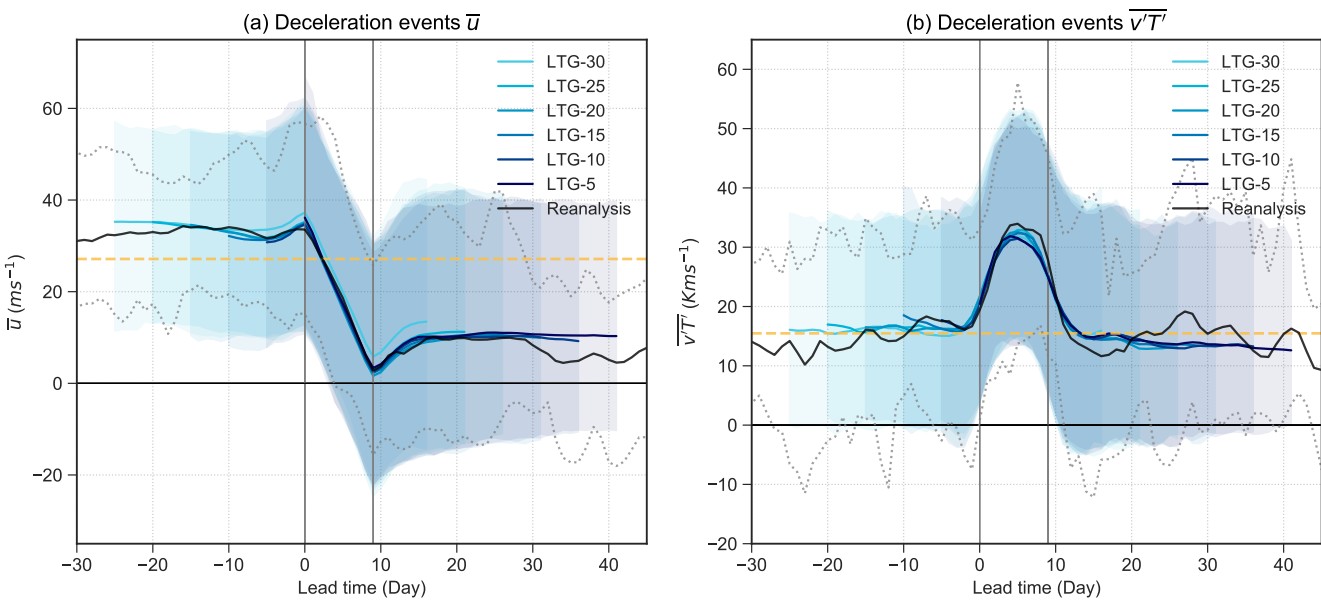

**Figure A3.** Like Figures 7a and 7e but excluding events with magnitude above the 90th percentile in the reanalysis composite.

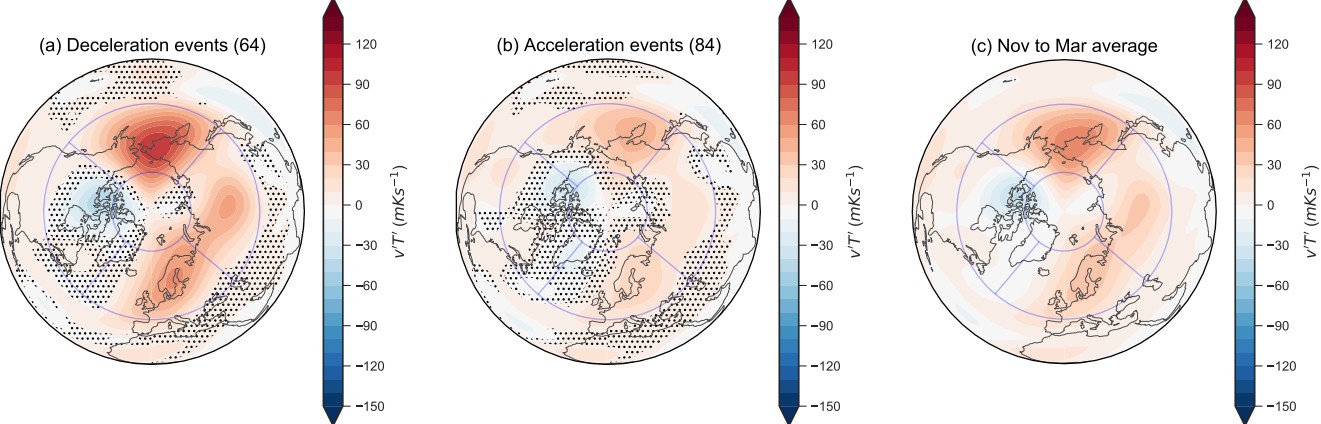

**Figure A4.** Composite $v'T'$ at 100 hPa averaged over day 0 to 9 (during the event window) of (a) Deceleration events, (b) Acceleration events and (c) Nov to Mar average in reanalysis. Blue lines mark the regions of investigation: Northern Europe ($40°$W - $50°$E), Siberia ($50°$E - $140°$E), North Pacific ($140°$E - $130°$W) and North America / Greenland ($130°$W - $40°$W). Numbers in parentheses indicate the number of events in the composite and unhatched regions in (a) and (b) indicate areas found to be significantly different from (c) using a t-test.

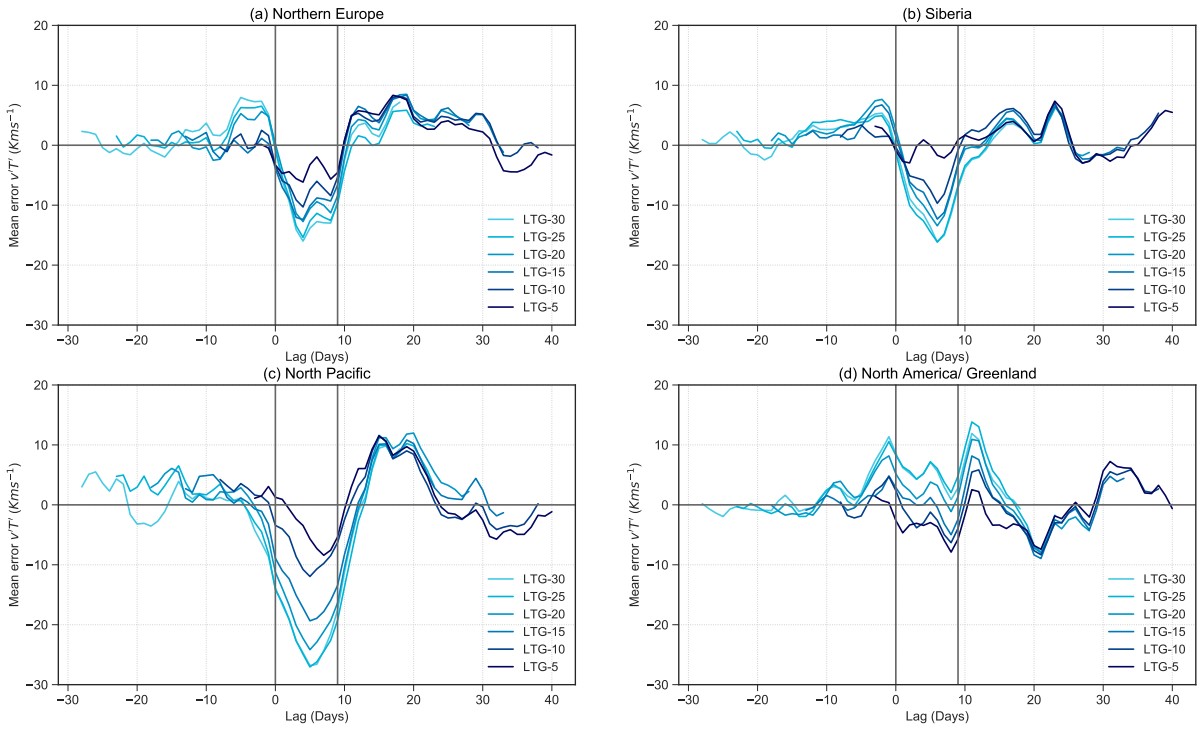

**Figure A5.** Mean error of the composite $v'T'$ at 100 hPa for deceleration events over (a) Northern Europe, (b) Siberia, (c) North Pacific and (d) North America / Greenland predicted by the hindcasts at different lead times.

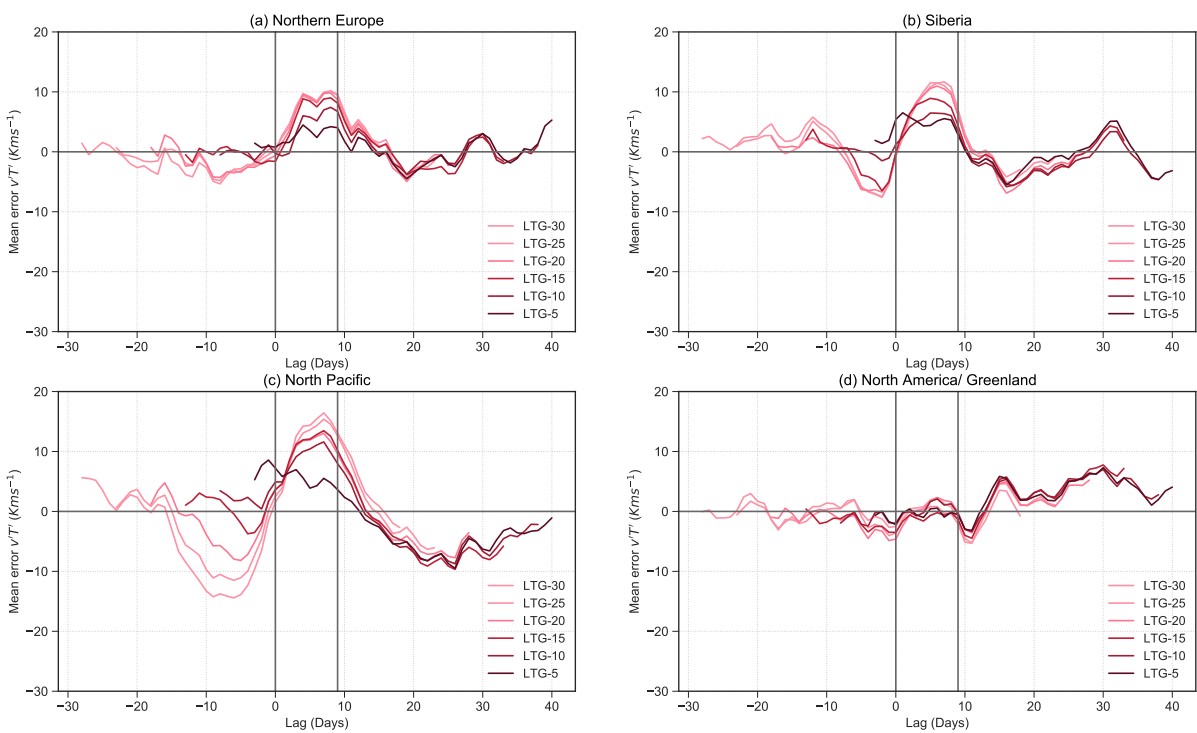

**Figure A6.** Same as Fig. A5 but for acceleration events.

*Author contributions.* R.W. and D.D. designed the study. R.W. performed the analysis, made the figures, and wrote the manuscript draft. All authors discussed the research and worked on revising the manuscript.

*Competing interests.* The authors declare no competing interests.

*Acknowledgements.* The authors thank Huw Davies, Hilla Afargan-Gerstman, and Bernat Jiménez-Esteve for helpful discussions on earlier versions of this research, and Ole Wulff for help with the S2S data. The authors would like to thank two anonymous reviewers and the co-editor Yang Zhang for their helpful comments throughout the revision of the manuscript. The work of R.W. is funded through ETH grant ETH-05 19-1 "How predictable are sudden stratospheric warming events?". The work of Z.W. is partially funded by the Swiss Data Science Center within the project *EXPECT* (C18-08). Support from the Swiss National Science Foundation through projects PP00P2_170523 to Z.W. and D.D. and PP00P2_198896 to D.D. is gratefully acknowledged. The ERA-Interim data was obtained from https://apps.ecmwf.int/datasets/ and the S2S data was obtained from https://apps.ecmwf.int/datasets/data/s2s-reforecasts-instantaneous-accum-ecmf/levtype=pl/type=cf/.

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
