# Peer review of "Differences in the Sub-seasonal Predictability of Extreme Stratospheric Events"

_Weather and Climate Dynamics, 2021_

## Author Comment (AC2)

**Response to the reviewers of wcd-2021-84: Differences in the Sub-seasonal Predictability of Extreme Stratospheric Events**

Dear Co-Editor Yang Zhang,

On behalf of all authors I would like to submit the revised version of the original article "Differences in the Sub-seasonal Predictability of Extreme Stratospheric Events" including an annotated manuscript and a modified version of the manuscript. During the revision, a number of changes have been made to the manuscript to satisfy the requests of the reviewers. Please find the summary of the changes and the detailed responses to the reviewers below.

Best regards,
Rachel Wu

We would like to thank both reviewers for their helpful comments and suggestions for our study. These have been included into the manuscript (see changes indicated in **bold** in the annotated manuscript). Please find below the detailed responses (in blue) to the reviewers' comments and suggestions. All line indications refer to the new (annotated) version of the manuscript unless specified. The main changes to the manuscript are listed here:

**Changes in Sections:**

- Section 2.2 *Skill measures* is now Section 2.3 and Section 2.3 *Definition of stratospheric events* is now Section 2.2

**Changes in Figures:**

- Figure 1: Separated into two panels, one for deceleration events, one for acceleration events

- Figure 3: Grouped every 2 LTGs together, reduced from 6 panels to 3 panels

- Figure 3, 6, 8, 9: now no longer distinguish weak and strong magnitude events

- Figure 4: Now showing 3 panels, one panel for each metric

- Figures 5 and 7: $\overline{u}_{yy}$ was in wrong units in the original manuscript, now corrected with the correct units

- Figure 6, 8, 9: Updated with $\overline{u}_{yy}$ averaged over day -10 to -1 instead of using $\overline{u}_{yy}$ at day 0

- Figure 7: we combined every two LTGs rather than showing all LTGs in the original manuscript

**Main topics of reviewer questions:**
We here list a few more general answers to major points that came up during the review.

1  **Chosen threshold for event definition vs. sample size:** We have carefully evaluated the chosen threshold for the defined acceleration and deceleration events with respect to the balance between the 'extremeness' of the events and the sample size. A more extreme threshold (i.e. a higher percentile) yields fewer and more extreme events, while it strongly reduces the sample size. Our goal is to have a larger sample size than what we obtain for limiting the sample to SSW and strong vortex events, but to keep the threshold sufficiently high to detect only events that can still be considered strong or even extreme, hence choosing the event threshold is a delicate balance. Several studies have been performed on the limited sample of SSW and strong vortex events, and given the limited amount of observational data it is difficult to understand the mechanisms for these events beyond case studies and simplified models. We therefore expand the sample to deceleration and acceleration events in order to better be able to find

commonalities between events, while still allowing for these events to be strong enough to exhibit many of the same mechanisms and characteristics that are observed for SSW and strong vortex events. It has to be noted that some of the strongest SSW events are possibly driven by mechanisms – such as resonance – that are highly nonlinear, and which may not be fully reproduced in the prediction system used here. The goal of our study is not to investigate the mechanisms for these very extreme events, but the basic ingredients such as wave driving and the background state that remain important ingredients worth investigating for all cases, and our study therefore aims to shed light on the nature of these events by balancing the sample size and the extreme nature of these events. We have elaborated further on this point in the detailed answers below.

2 **Nonlinear nature of stratospheric events:** As noted in point 1 above, the most extreme SSW events may exhibit different and/or additional mechanisms that are highly nonlinear. For example, some of the most extreme SSW events have been suggested to be driven by resonance. However, all deceleration events including strong SSWs require a minimum threshold of wave flux input into the stratosphere, and furthermore the stratospheric background state is often found to be critical for wave amplification. Therefore, these two ingredients are two clear starting points for our study, i.e. to investigate if the model is able to reproduce the wave flux and the stratospheric state in order to be able to reproduce stratospheric events. By no means do we intend to suggest that stratospheric events are driven by purely linear mechanisms, but it is rather the deviation from linear relationships that we are interested in. Nevertheless, we find relationships that are close to linear in terms of their predictability (rather than their dynamics), e.g. the relationship between the CRPS and the wind change (Figure 3). We have elaborated further on this point in the detailed answers below.

3 $\overline{u}_{yy}$ **as a measure of the background state:** We demonstrate with scatter plots in Figures 6-10 in this document that $\overline{u}_{yy}$ correlates well with $\overline{u}_{zz}$ (the third term of $\overline{q}_y$), and that $\overline{u}_{yy}$ correlates well with $\overline{q}_y$. In Lines 183-185 in the manuscript, " Other than being a reasonable indicator for the refractive index, $\overline{u}_{yy}$ is a measure of the sharpness of the edge of the stratospheric polar vortex, thus also a measure of the strength of the initial vortex state." Thus, in the light of the scatter plots and correlation we include in this response, we decided to keep our interpretation of $\overline{u}_{yy}$ as a measure of the background state of the vortex, and a reasonable proxy for the refractive index, for the region we consider. To filter out high frequency variations in $\overline{u}_{yy}$, we have modified our $\overline{u}_{yy}$ index by using a 10-day averaged value of $\overline{u}_{yy}$ at day -10 to -1 instead of using the value of $\overline{u}_{yy}$ at day 0 that was originally used in the study.

**Reviewer 1**

**General Comments**

The manuscript addresses an important gap in the subseasonal-to-seasonal (S2S) community - an investigation of how predictable rapid acceleration and deceleration polar vortex events are in the ECMWF subseasonal forecasting system. Quantifying this predictability is important, as changes in the strength of the Northern Hemisphere stratospheric polar vortex typically precede changes in winter weather regimes in the troposphere. The authors find that, while the ECMWF performs well in terms of the driving mechanisms for these acceleration/deceleration events, it cannot capture the magnitude of the most extreme events, a finding common to other prediction systems. This discrepancy in magnitude is especially true for the wave fluxes, which are underestimated in the model. Altogether, the analysis of the model and comparisons with reanalysis is done generally well, and the authors have identified a couple of key metrics that could be assessed for these events. These two metrics - meridional heat flux and a proxy for the index of refraction - could be useful in future assessments of subseasonal forecasting systems and their stratosphere-troposphere coupling mechanisms. The statistics shown are valid and comprehensive, though admittedly numerous and could be streamlined. I think that the conclusions follow the analyses conducted, though a bit more on the mechanistic framework and some more spatial-dependent analyses could help the paper. As such, I am suggesting that the work undergo major revisions before acceptance.

| $\overline{v'T'}$ ⟍ $\overline{u}_{yy}$ | -30 to -21 | -20 to -11 | -10 to -1 | 0 to 9 |
|---|---|---|---|---|
| 0 to 9 | 0.47/0.00 | 0.44/0.00 | 0.30/0.02 | - |

| $\overline{v'T'}$ ⟍ $\overline{u}_{yy}$ | -30 to -21 | -20 to -11 | -10 to -1 | 0 to 9 |
|---|---|---|---|---|
| 0 to 9 | - | - | - | - |

Figure 1: Correlation of 10-day integrated sum of $\overline{v'T'}$ over day 0 to 9 and the 10-day averaged $\overline{u}_{yy}$ index at different lags, e.g. -30 to -21 denotes 30 to 21 days before the start of the events. The Pearson coefficient (first value) and p-value (second value) are only shown for significant correlations. The top table shows the values for deceleration events and the bottom table for acceleration events.

Thank you for your insightful evaluation of our manuscript. We are responding to your detailed comments below.

**Specific Comments**

- **Interdependence of Refractive Index and Wave Forcing.** The authors examine mechanisms and drivers that could explain strong acceleration and deceleration events. To do this, they have examined the index of refraction and meridional heat flux. However, the authors indirectly treat these two metrics as independent and look at their evolution separately. In fact, the authors treat the index of refraction as a measure of the "background state of the stratosphere" (Line 245). However, these two variables are a function of each other. While initially the refractive index may facilitate wave propagation, the breaking of waves in the stratosphere and the changes in the zonal winds and heating profiles caused by these breaking waves will alter the refractive index, which in turn influences future wave breaks. So, it is hard to keep the two metrics completely separate. Have the authors considered this interdependence and thought of ways to address it? For example, if a model poorly handles wave fluxes 25-30 days before an observed event, can we actually use the simulated index of refraction to assess its prediction of an event?

  Thank you for your comment. We agree that the two selected indices, i.e. the refractive index and the meridional heat flux, are related to each other. To address the problem of interdependence, we revisited the definition of the metrics and considered the averaging window at different time lags for the metrics, to better separate the indices.

  We compare the correlations of 10-day averaged $\overline{u}_{yy}$ index at different lags with the 10-day integrated sum of $\overline{v'T'}$ over day 0 to 9 for both deceleration and acceleration events at different lags in Figure 1. In the top table, for deceleration events, we find significant positive correlations in $\overline{u}_{yy}$ with the integrated $\overline{v'T'}$ over day 0 to 9 whenever $\overline{u}_{yy}$ leads $\overline{v'T'}$, i.e. for $\overline{u}_{yy}$ averaged over day -30 to -21, day -20 to -11 and day -10 to -1. In the bottom table, for acceleration events, no significant correlation is found between $\overline{u}_{yy}$ and the integrated $\overline{v'T'}$ over day 0 to 9.

  As such, we agree that we should not treat the two indices as independent for deceleration events. To minimise the dependence of the two indices, we choose the time lag that exhibits the lowest correlation between the two metrics, i.e. day -10 to -1 for $\overline{u}_{yy}$, as the new averaging window for the $\overline{u}_{yy}$ metric. We have also added Lines 189-194 in the manuscript to clarify that these indices should be treated with care and that the two metrics are not fully independent.

- **Spatial Analyses.** The manuscript studies all events and their forcings in a zonal-mean framework. That approach is a classical way to look at stratosphere-troposphere coupling, but emerging evidence points to the importance of polar vortex morphology and tropospheric source regions of waves for understanding circulation anomalies in the troposphere and stratosphere. As such, spatial distributions of meridional heat flux (at a given isobaric level or even as a cross-section) could be very informative to understand whether the models initiate the waves in the right places. For example, climatologically, vertical wave propagation has two major hotspots during boreal winter: (a) Siberia and (b) Scandinavia

/ Northern Europe. However, other forecasting systems possess biases on where these hotspots are because of their representation of planetary-scale waves. How does the ECMWF perform in this context, and specifically during strong acceleration or deceleration events? Is one region better represented than the other? Also, what about the morphology of the stratospheric polar vortex? How is that different in the lead up to strong and weak acceleration events, and could that be a predictive element? I am offering two suggestions here, but others are possible. My main point is that I would like to see more multi-dimensional analyses in addition to the zonal-mean metrics (which are important!).

Thank you for your comments and suggestion. We agree that regional analysis of wave activity would be interesting and a nice addition to our results. As an extra analysis, we have separated the reanalysis data of meridional heat flux into contributions from four regions that are selected based on the existing literature and on our own analysis. Specifically, based on the meridional heat flux composite of deceleration events (Fig. 2a), we divided the 45-75°N latitude region equally into four regions as indicated in Figure 2: (a) Northern Europe (40°W - 50°E), (b) Siberia (50°E - 140°E), (c) North Pacific (140°E - 130°W) and (d) North America/ Greenland (130°W - 40°W). Comparing the composite of deceleration events to that of the Nov to Mar average, we see positive heat flux anomalies in three regions, namely, Northern Europe, Siberia and the North Pacific, and negative heat flux anomalies in the region North America/ Greenland during deceleration events. The composite for acceleration events also shows similar patterns. Thus, we choose to average the heat flux over the same four regions for both deceleration and acceleration events to examine the predictability of the wave activity captured by the model at different lead times.

[Figure]

Figure 2: Composite $v'T'$ at 100 hPa averaged over day 0 to 9 (during the event window) of (a) Deceleration events, (b) Acceleration events and (c) Nov to Mar average in reanalysis. Blue lines mark the regions of investigation. Northern Europe (40°W - 50°E), Siberia (50°E - 140°E), North Pacific (140°E - 130°W) and North America / Greenland (130°W - 40°W). Numbers in bracket indicate number of events in the composite and unhatched regions in (a) and (b) indicate statistically significant different from (c) by a t-test.

[Figure]

Figure 3: Mean error of the composite $v'T'$ at 100 hPa for deceleration events over (a) Northern Europe, (b) Siberia, (c) North Pacific and (d) North America / Greenland predicted by the hindcasts at different lead times.

[Figure]

Figure 4: Same as Fig. 3 but for acceleration events.

Fig. 3 shows the mean error of the averaged $v'T'$ for the four regions for the deceleration events in the hindcast with respect to the reanalysis. The mean error is largest in the North Pacific, while it is of similar magnitude for Northern Europe and Siberia, with larger errors for longer lead times. The model generally underestimates $v'T'$ during the event window in these three regions. The error is comparably smallest for the North America/ Greenland region. Since the averaged heat flux is negative in the region, the positive mean errors at long lead time indicate that the averaged $v'T'$ in the model of the region is not as negative as that in reanalysis. At short lead time, the mean errors for North America/ Greenland become negative, indicating that the $v'T'$ for the region in the model becomes more negative than in reanalysis.

The same analysis is done for the acceleration events ( Fig. 4). The mean error in the North America/ Greenland region is close to zero for all LTGs, while the model overestimates the $v'T'$ for the North Pacific, Northern Europe and Siberia. The error in the North Pacific is larger than for Northern Europe and Siberia, as observed for deceleration events. However, the errors in the North Pacific for acceleration events are more comparable in magnitude to Northern Europe and Siberia than for deceleration events.

To sum up, $v'T'$ in the North Pacific contributes more than the other regions in terms of the errors in $v'T'$ and the North America/ Greenland region contributes less. However, all regions contribute to the $v'T'$ errors for the deceleration events and all regions except the North America/ Greenland regions for the acceleration events. A more detailed analysis on the regional origin of errors will have to be performed in the future.

To describe the new results in the manuscript, we have added Lines 406-410: 'We further investigate the regional origin of the $\overline{v'T'}$ errors by dividing up the regions of heat flux origin into Northern Europe, Siberia, North Pacific and North America/ Greenland (Fig. A4). All regions contribute to the errors, with the largest contribution coming from the North Pacific, and smaller contributions from Northern Europe and Siberia (Fig. A5 and A6). Additional analysis is needed to further understand the origin of the $\overline{v'T'}$ errors.' We have also added the figures into the Appendix of the manuscript as Fig. A4 to A6 and the description to Fig. A4 in Lines 467-476.

- **More Justification for Choice of Events.** The authors provide definitions for their strong and weak magnitude events as being above their respective 60th percentiles. However, I am unsure why this percentile is chosen other than that threshold is used in other works. In fact, I do not consider the 60th percentile as "extreme" as the title of the manuscript indicates. I would like the authors to provide more details on the choice of this threshold and also how sensitive their analyses and conclusions would be if the value was shifted to the 75th or 80th percentiles.

[Please see also major point 1 at the beginning of the reviewer response] Thank you for your comment. The chosen threshold is based on obtaining strong events while also having a large enough sample that is larger than the limited sample of SSW or strong vortex events. We have made an effort to put our chosen threshold in context by comparing with previous studies that define similar events. For example, the threshold that we use for strong magnitude events has a similar magnitude as the definition used in Birner and Albers (2017), who classified *sudden stratospheric deceleration events*. We have, therefore, re-framed our event definition in order to better justify our choice of threshold for events and to what extent the classified events can be considered extreme events.

In Birner and Albers (2017), the threshold is chosen based on the standard deviation of the deseasonalised daily zonal mean zonal wind of the time period they considered, where the threshold for a 10-day event is defined by multiplying the standard deviation of the daily value by 10 days. The 10-day threshold they use is 20 ms$^{-1}$ over 10 days, which corresponds to a daily zonal mean zonal wind change of around $2.2\sigma$ per day, which is equivalent to a daily wind change of around the 98th percentile. Following Birner and Albers (2017), we compute the standard deviation of the time series of the deseasonalised zonal mean zonal wind of our time period of consideration, which is around 1 $ms^{-1}/day$. Therefore, our definition of strong acceleration and deceleration events of 16.9 ms$^{-1}$ over 10 days and 24.6 ms$^{-1}$ over 10 days corresponds to daily wind changes between the 95th and 99th percentile values ($1.69\sigma$ and $2.46\sigma$) in NH Nov-Mar. Therefore, we keep our original definition of events. We have adapted Lines 103-125 of the manuscript to better explain our choice of threshold.

Having revisited the figures in the manuscript, in most of the figures of the original manuscript, we

distinguish between weak and strong magnitude events. We found that this distinction is not needed for most of our conclusions, and we draw conclusions from most of the figures using the entire event magnitude spectrum, i.e. considering all events together rather than just the strong magnitude events. However, in Figure 4 and 7, the strong magnitude events need to be distinguished to convey the messages from the figures. As such, other than Figure 4 and 7, we have modified all the other figures to not indicate the difference between weak and strong magnitude events.

As we now no longer distinguish strong magnitude events in most of the figures, it will only affect Figure 4 and 7 if a higher percentile is used as the threshold of the events. As a sensitivity test, we repeated the same analyses but using the 80th percentile. The results still apply for Figure 4 and 7. However, when using the 80th percentile as the threshold, much fewer events are detected and the number of identified strong magnitude events become comparable to the number of SSW events. We have therefore kept the original percentile definition and elaborated on it in Lines 116-125 in the manuscript.

- **Complexity of Figure 7.** I understand the motivation of looking at multiple lead times and ensembles when studying these different events and comparing their features to reanalysis. However, Figure 7 has seven differently colored lines (six of which are different shades of blue), two different line styles, and six colors of shading per panel. I found it difficult to differentiate the different blue colored lines, especially since many of them overlap each other. I think the authors should consider simplifying these figures by, for example, reducing the quantity of lines. Since we already know that the models improve with shorter lead times from the other previous analyses, can the same message come across with just LTG-25, LTG-10, and LTG-5? Are all the shading colors needed? Again, I am thinking of ways of making this figure more accessible and cleaner without losing its meaning.

  Thank you for the comment. We have modified Figure 7 by plotting the average of every two LTGs to reduce the number of colours and lines to make the figure cleaner.

**Technical Corrections**

- **Lines 1-2.** The phrase "associated with an anomalously weak or strong polar vortex" is oddly placed. Please consider removing this phrase.

  Thanks, phrase removed.

- **Lines 10.** Please add a semicolon after "behaviour."

  Thanks, change made.

- **Lines 10-11.** The wording following "that is" reads awkwardly. Please consider revising.

  Thank you for the comment. The sentence is now reformulated in Lines 8-10.

- **Lines 34-35.** How does the strong latitudinal temperature gradient drive radiative cooling in the stratosphere? Isn't the radiative cooling a function of the (lack of) solar insolation during winter months?

  Thank you for this comment, the sentence was misleading. This sentence has been modified and combined with the following sentence and now reads "On the other hand, when wave activity is weak and the SPV is relatively undisturbed, the vortex strengthens on radiative timescales (Limpasuvan et al., 2005; Hitchcock and Shepherd, 2013)." in Lines 33-34.

- **Lines 39.** Please add "Major" before "SSW events."

  Thanks, change made.

- **Figure 1.** I suggest that the authors break this figure into two panels: one for the deceleration/SSW events and the other for the acceleration/strong vortex events. As presented, the one plot has a lot of information and is too cluttered to understand fully. Moreover, is **Line 197** correct? When I examine the figure, I see the blue line (median for deceleration events) higher than red line, indicating a higher magnitude error for deceleration events, not the other way around. Maybe it is just hard to see in the figure (for me), but could the authors check this and perhaps explicitly state the values of the medians just to make sure?

Thank you for the comment. We have now split Figure 1 into two panels as suggested. Thank you for spotting the error at Line 197 in the original manuscript. The median for deceleration events is larger than acceleration events in LTG-5, which indicates that mean error for deceleration events is higher than acceleration events for all lead times. We have now replaced the sentence with '..., we also find that deceleration events are associated with larger errors than acceleration events at all lead times.' in Line 218-219.

- **Line 203.** Please add "wind changes" after "magnitude" to make clear what the magnitude represents.

  Thanks, change made (Line 224).

- **Figure 2.** In the caption, please change "brackets" to "parentheses."

  Thanks, change made.

- **Lines 213-216.** I read this sentence several times, and I still do not understand what it is saying about the gray diagonal line. Please consider rewriting.

  Thank you for pointing this out. We have added the following sentence on Lines 156-159 in Section 2.3: "As the CRPS is given by the difference between the predicted and observed distribution, if all ensemble members in a hindcast predict an event magnitude of 0 $ms^{-1}$, i.e. close to a climatological state where the wind stays relatively constant during a 10-day window, the CRPS of this hindcast will be equal to the observed event magnitude itself.". We here aim to explain how the CRPS of a hindcast that predicts a climatological state will be equal to the event magnitude of the actual event. Then, in Lines 233-235 of the Results section, we shortened our description on the grey diagonal line and stated directly how the diagonal line is used as a reference to compare the skill of the data points to a climatological forecast.

- **Lines 228-229.** This line starting with "For instance" is a fragment and should be corrected.

  Thank you for catching this. We have modified the line to "Some events, for instance, the two extreme SSW events with magnitudes of over 60 ms$^{-1}$ (marked by yellow stars in Fig. 3), retain a large CRPS and deviate from the linear fit in the direction of the diagonal line." in Lines 245-247.

**Reviewer 2**

**General Comments**

This paper examines predictability of wind deceleration and acceleration events using the ensemble hindcasts of the ECMWF for the period of 1998-2018. The variability and predictability of those events are examined according to the magnitude change of the zonal-mean zonal winds, its meridional curvature at 60N, 10 hPa and eddy heat fluxes in the lower stratosphere. It is found that the model can reasonably predict the acceleration events but unable to reproduce extremely deceleration events, which effectively the SSWs. The inability of the model to produce SSWs is linked to weaker-than-observed eddy heat fluxes in the lower stratosphere within the same 10-day interval.

The evaluation of the statistical representation of acceleration and deceleration events are interesting, e.g. the model continues to underestimate the long tails associated with deceleration events, even at short lead times; how the distributions of various quantities compared with reanalysis data sets. I however have major concern in terms of the dynamical reasoning. See comments below for details.

Thank you for your comments and insightful evaluation of our manuscript. Below are point-by-point responses, and we also added further discussion in the revised manuscript in response to your comments.

**Major Comments**

- The mechanisms that the authors identified are entirely consistent with the linear theory, which is adequate in explaining the climatological behaviour of stratosphere wave mean-flow interaction and

polar vortex variability, but not sufficient in explaining the SSWs. Thus, the title of the paper does not match its content or key results.

[Please see also major point 2 at the beginning of the reviewer response] Thank you for your comment. We fully agree that linear theory is not sufficient to explain SSW events and that processes that are described by nonlinear theory, such as resonance, would be needed to – in particular – fully explain very strong SSW events. We do not aim to, in our paper, to explain SSWs with linear theory. Instead, our aim is to trace the sources of the predictability of stratospheric extreme wind events in the model. As a first approximation, we investigate deviations from linear relationships between the predictability of events and their magnitude (e.g. Figures 3 and 9) and between precursors and event magnitude (Figure 6). The finding is that the stratospheric events follow linear relationships to some degree, but that especially extreme events – unsurprisingly given their nonlinear dynamics – deviate from the linear relationship.

The model in general captures the linear part of the wave activity forcing well for most events except for the strong magnitude events, implying that the model shows an inability to capture very strong wave activity, which might be related to the inability of the model to capture nonlinear processes. As such, (please also refer to the response to the next comment) we think our results are important for extreme stratospheric events, since the strong magnitude events we identified have magnitudes that are comparable to extreme stratospheric events. For more justification of the 'extremeness' of the events please see the answer to reviewer 1, point 3 ('More Justification for Choice of Events'). We have therefore decided to keep the title of our paper.

- The results presented shade little new insight onto the predictability of extreme stratospheric events, i.e. SSWs. This is mainly because the authors use upper and lower 60th percentiles of negative (or positive) deltaU within a 10-day window to define the deceleration (or acceleration) events, which is not the standard measure of extreme events. For instance, a normal distribution can approximately capture the 60 percentiles of generalized extreme value (GEV) distribution, but it would fail to model the long tails of the GEV, which normally corresponds to bottom or top 1-5 percentiles of a distribution. Thus, including small-magnitude events will result in better statistics but potentially hides the responsible mechanisms for the extreme events because the statistics provided by the 60 percentiles of a population is not representative of its extreme values.

[Please see also major points 1 and 2 at the beginning of the reviewer response] Thank you for the comment. We would like to first clarify that by the term extreme stratospheric events, we are referring to both strong vortex events and SSW events (see Section 2.2 for more detailed definitions), which are commonly studied in predictability studies, as well as strong deceleration and acceleration events, which have a strong overlap with SSW and strong vortex events.

We are aware that the definition of using the 60th percentiles of deltaU within a 10-day window might not be considered extreme. However, when we present our definition of extreme events in terms of wind change per day, our definition for deceleration and acceleration events correspond to wind change within the 95th and 99th percentiles, respectively, per day. (For more justification of the 'extremeness' of the events please see the answer to reviewer 1, point 3 ('More Justification for Choice of Events').) Our definition of the strong deceleration events is comparable to the definition of sudden stratospheric deceleration events or alternative definition of SSWs based on zonal-mean zonal wind tendency in (e.g. Birner and Albers, 2017; de la Cámara et al., 2019; Kim et al., 2017), where the threshold they use corresponds to around daily zonal mean zonal wind change of around 2.2 standard deviations per day and to a daily wind change of at around the 98th percentile.

Following the definition of Birner and Albers (2017), we have computed the standard deviation of the time series of the deseasonalised zonal mean zonal wind, which is around $1 \ ms^{-1}/day$. Our definition of strong acceleration and deceleration events, which is defined as events with wind change of 1.69 $ms^{-1}/day$ and 2.46 $ms^{-1}/day$ within a 10-day event window respectively, corresponds to 1.69 times of the wind changes of 95th and 99th percentile ($1.69\sigma$ and $2.46\sigma$) in NH Nov-Mar. We have added Lines 112-125 in the manuscript to better elaborate and justify our choice of event definitions.

Furthermore, we agree that the strongest SSW events, although part of the sample presented here, are not fully captured in terms of predictability. In fact, we show that the strongest SSW events, e.g. the split events in 2009 and 2018, are very poorly captured in the prediction system (e.g. Fig. 3 in the

manuscript). We are not able to investigate these events in detail in this study due to their exceptional nature, and we have in this study rather focused on studying the general ingredients of predictability for wave acceleration and deceleration events. However, the extreme nonlinear events that are highly unpredictable are a highly interesting further topic to explore in a future study.

- The deceleration and acceleration events appear to include high frequency variability (i.e. < 5 days), the effect is readily seen in Figure A1. The authors need to either justify the extent to which the effects of these high-frequency waves on the polar vortex variability in relation to the SSWs or applying a lowpass filter to the 6-hourly data so that the variation within the 10-day window is truly relevant to extreme stratospheric events.

  Thank you for this comment. We agree it is undesirable to include high-frequency variability within the 10-day event window. We have actually taken this point into consideration when we identify the events. It was, however, not included in the original manuscript. Thank you for pointing this out.

  We have now included Lines 103-105, 'We also impose a criterion that the ratio of the maximum difference in between the maximum and minimum wind speed occuring during the 10-day event window has to be less than 1.2, to filter out high frequency variations.', to describe that a criterion is imposed when identifying events to avoid picking up events that include high-frequency variability. We have also included Harnik (2009) as a reference in Line 453 to comment on the effect of high-frequency variability on the polar vortex.

- The SSWs are known to involve nonlinear processes such as wave breaking, resonance, and internal wave reflection, some of which the model may have failed to capture. For instance, erosion and filamentation due to wave breaking can increase meridional curvature as well as enhance zonal winds at polar vortex edge via PV sharpening. Thus, wave forcing from below does not always result in a weaker polar vortex within a 10-day time window. As such, the meridional curvature term uyy is not a good measure of waveguide.

  [Please see also major point 2 on first page of reviewer response] Thank you for your comment. We fully agree that SSWs involve nonlinear processes. In our results, the model however shows limitations in producing extreme values of the eddy heat flux, which already indicates that the model does not fully capture processes such as resonance and internal wave reflection. This is, however, beyond the scope of our study.

  We also agree that wave forcing from below does not always result in a weaker polar vortex, and that the vortex can also be strengthened via PV sharpening. However, for longer timescales, i.e. the 10-day averages that are employed in our study, (e.g. Figure 5 and 6 in the manuscript), the 10-day integrated eddy heat flux during the 10-day event window has a strong positive correlation with the wind deceleration in the window, yielding an overall clear response. Hence, if we consider the integrated eddy heat flux in this time window (i.e. day 0 to 9), the wave forcing from below will most of the time result in a weaker polar vortex.

  In the light of your comment, we have added Lines 449-453 in the Discussion, 'For example, the ability of the model to capture the nonlinear dynamics, which are known to be relevant to SSWs with strong magnitude, has not been explored in this study. These nonlinear processes include the complex behavior of wave breaking, which depending on its exact location and temporal variability can have different effects on the polar vortex, for instance, high frequency wave activity can strengthen the polar vortex rather than weakening it (**?**). As such, ...'.

- A few multi-panel figures are too complicated and some of the panels are redundant. See specific comment below.

  Thank you for this comment, we have now simplified several of the figures in the revised manuscript following the comments by both reviewers. We have added a summary of the changes at the beginning of the reviewer response, please see above.

**Specific Comments**

- Line 4, page 1, delete "limit".

Thanks, change made.

- Line 10, page 1, "in a close to linear relationship", it may not be appropriate to study extreme events using linear relationship.

  Thank you for the comment. In this case, we are describing the predictability of the events and comparing how closely the relationships between event magnitude and precursors, and between their predictability in the model resemble a linear relationship. By no means do we intend to imply that this relationship should be linear, or that the dynamics of the events should be linear. The regression line is used as a reference to compare the relationship between the wind change and the integrated eddy heat flux and the averaged $\overline{u}_{yy}$. We have now clarified this in Lines 196-199 in the manuscript.

- Line 13, page 1, "wave activity pulses", I do not think that the authors studied wave activity pulses. The exact quantity studied is v'T' averaged within a 10-day window, which can contain only a part of wave pulse or multiple high-frequency wave pulses.

  Thank you for this comment. We have replaced "wave activity pulses" with "wave activity fluxes" in Lines 12 and 14.

- Lines 35 and 45, page 2, polar vortex can be strengthened via wave breaking and PV sharpening as well.

  Both sentences have been adapted, see Lines 33-34 and 44.

- Line 59-60, page 3, very good point Re initial stratospheric conditions, but the authors did not study this factor in the rest of the paper. Consider rephrase or remove the sentence.

  The second half of the sentence has been rephrased, in Lines 58-59, "..., suggesting that other factors, e.g., the background state of the stratosphere, might be important for successful predictions of SSWs.", as we investigate the predictability of the background state of the stratosphere in our study.

- Lines 123-134, page 5, using a fixed 10-day moving window to define the acceleration and deceleration events is problematic as it cannot properly differentiate high and low frequency variability thereby wave mean flow interaction. Harnik (2009) demonstrated that low frequency wave activity slows down the zonal winds while transient, high frequency wave pulses act to enhance the polar vortex.

  Thank you for this comment. One reason for using a 10-day event window, apart from having a consistent time window for defining wind changes, is to filter out high frequency signals (see Lines 103-111 in the manuscript). While we are aware that high frequency wave pulses also have an effect on enhancing the polar vortex, high frequency wave pulses are not the focus of this paper. To account for this finding, we have included a sentence about the effect of high frequency pulses into the manuscript in Line 453, citing the reference by Harnik (2009). In addition, we have evaluated a wide range of window widths as a sensitivity test. Figure A1, panel (a) shows that there is no major difference between acceleration and deceleration events in terms of their duration.

- Lines 141 -155, it is better to condense those roles/conditions and put them into Table 1. Also, the sample size for each subgroup in reanalysis are too small to establish robust statistics or to understand the relevant mechanisms. For instance, a strengthening of a polar vortex can be due to reduced upward wave forcing, PV sharpening via wave breaking, and/or enhanced meridional temperature gradient. It is nearly impossible to differentiate these causes merely based on 25 events.

  Thank you for your suggestion. We agree that combining those definitions into Table 1 will make it clearer for the reader. We have added the definitions as an extra column to Table 1. We have, however, kept the text that explains the criteria in the manuscript as we think it might allow readers to better follow the manuscript.

  About the sample size, we agree that we should not identify or differentiate causes and the mechanisms based on the small sample size from the reanalysis data. The goal of this study is to investigate and compare the predictability of stratospheric wind changes, and to identify possible model biases. We extend the definition from just SSW and strong vortex events, which have often been studied in terms of their predictability despite small sample sizes, to acceleration and deceleration events, which have a larger sample size. A further goal of the study is to give hints at the potential mechanisms that may not be sufficiently represented in the prediction model in order to give an accurate prediction. We therefore

evaluate whether the model captures the mechanisms as suggested from the literature and whether the model has biases in predictors that are known from the literature. The investigation, and thereby the sample size, is a balance between obtaining a sufficient number of events (that is larger than the observed record for SSW and strong vortex events) and the threshold for defining these events. We found that the chosen threshold gives us a number of events that allows for an investigation of the mechanisms, while still having events that are strong enough to compare to e.g. SSW events. [please see also major point 1 at the beginning of the reviewer response]

- Line 153, "the chosen threshold . . . ", at which pressure level and latitude?

  Thank you for pointing this out. We have replaced "the chosen threshold . . . " with the information on pressure level and latitude as "the chosen threshold value at $60°$N and 10 hPa is 41.2 m/s" in Line 138.

- Lines 169-170, I am not convinced that the meridional curvature at 55-75N, 10 hPa is a good measure of refractive index for stationary planetary waves. The climatological EP fluxes at this latitude band and height location are mainly upward, suggesting the dominant role of vertically propagating Rossby waves. This also implies the important role of the vertical component of the refractive index. It is the first time for me to read that the third term in the equation (5) is highly corrected with meridional curvature term at 55-75N, 10 hPa for the entire winter period from November to March. I would appreciate if the authors can demonstrate the correlation using scatterplots of the reanalysis data also the hindcasts using the 10-day window.

  Thank you for your comment. We revisited the correlations between the second term and third term of the meridional PV gradient, and the second term of the meridional PV gradient with the meridional PV gradient. In Figures 5 to 8 in this document, we show scatter plots of the second and the third term of the meridional PV gradient at 55-75N, 10hPa, and the second term with the meridional PV gradient for the reanalysis data.

[Figure]

Figure 5: Scatter plots of the daily values of the third term ($\overline{u}_{zz}$) and the second term ($\overline{u}_{yy}$) of the meridional PV gradient for deceleration events during different periods around the events for reanalysis data. Each data point corresponds to a daily value.

In each panel of the scatter plots, we plot the daily values of the corresponding quantity during the specified period. For instance, for days -40 to 40, we plot the daily values of all 81 days against the corresponding quantity of interest on the same day. We compute the Pearson correlation for each scatter plot, the computed coefficient and p-value are included in the legend of each plot.

In Figure 9 in this document, we summarise the computed correlation of all the scatter plots. We show that at 55-75$°$N, 10 hPa, where $\overline{u}_{yy}$, $\overline{u}_{zz}$, $\overline{q}_y$ are averaged, the second term of the meridional PV gradient correlates well with the third term. The second term of the meridional PV gradient is also highly correlated with the meridional PV gradient. Therefore, we think that for the specific region and level we are looking at, i.e. 55-75$°$N, 10 hPa, the second term of the meridional PV gradient, $\overline{u}_{yy}$, is a good approximation for the meridional PV gradient and we continue to use $\overline{u}_{yy}$ to approximate the meridional PV gradient.

[Figure]

Figure 6: Same as Figure 5 but for acceleration events.

[Figure]

Figure 7: Scatter plots of the daily values of the second term of the meridional PV gradient ($\overline{u}_{yy}$) and the meridional PV gradient ($\overline{q}_y$) for different periods around deceleration events for reanalysis data.

[Figure]

Figure 8: Same as Figure 7 but for acceleration events.

| $corr(\overline{u}_{zz}, \overline{u}_{yy})$ | -40 to 40 | -10 to -1 | 0 to 9 |
|---|---|---|---|
| Deceleration events | 0.74/0.00 | 0.69/0.00 | 0.78/0.00 |
| Acceleration events | 0.75/0.00 | 0.73/0.00 | 0.79/0.00 |

| $corr(\overline{q}_y, \overline{u}_{yy})$ | -40 to 40 | -10 to -1 | 0 to 9 |
|---|---|---|---|
| Deceleration events | 0.95/0.00 | 0.94/0.00 | 0.97/0.00 |
| Acceleration events | 0.95/0.00 | 0.96/0.00 | 0.96/0.00 |

Figure 9: Table summarising the Pearson correlation coefficient ($r$) and p-value ($p$) for Figures 5 to 8. Values are written in the format $r/p$.

- Lines 173-175, are the $\overline{u}_{yy}$ at 55-75N, 10 hPa and v'T' at 45-75N, 100 hPa both calculated using the same 10-day window as well?

  Thank you for pointing this out. In the original manuscript, we take $\overline{u}_{yy}$ at 55-75°N, 10 hPa at day 0 and $\overline{v'T'}$ 45-75°N, 100 hPa integrated over days 0 to 9. However, having considered the comments from both reviewers, we have modified to average $\overline{u}_{yy}$ to be computed over days -10 to -1 instead.

- Figure 3, without losing any information, a, b, and c can be combined into one panel. Also, panels d and e can be combined into one panel.

  Thank you for the suggestion. We have now combined the LTG-30 and 25, LTG-20 and 15, LTG-15 and 5, respectively into a total of 3 panels.

- Figure 4, if all the dashed lines were removed, would it lose any of the key information that the authors want to deliver regarding extreme stratospheric events?

  Thank you for the suggestion. We plotted the dashed lines which refer to the errors corresponding to weak magnitude events in the Figure as a comparison to the strong magnitude events. By comparing the predictability of weak and strong magnitude events, we want to bring out the reason for why the events with stronger magnitude are less predictable. We have also now modified the figure by showing one metric in each panel, comparing the metrics for acceleration and deceleration events of weak and strong magnitude.

- Figure 6, because the focus of the paper is on the predictability, only panels (b) and (e) are worth shown.

  Thank you for the suggestion. As mentioned in Line 292-294 in the manuscript, on average SSWs occur towards the end of the event window, i.e. we identified most SSW events around day 6. Therefore, we still find the panels for lags of days -10 to -1 and days 0 to 9 relevant for extreme events, e.g. SSWs. And as the background state may also matter, we would also need panels (a) and (d), which we have also now modified by taking the day -10 to -1 average for $\overline{u}_{yy}$.

- Figure 7, the temporal evolution of $\overline{u}_{yy}$ is almost identical to that of u itself, this implies that $\overline{u}_{yy}$ at the polar vortex edge is not necessarily a good measure of refractive index as it contains the same high frequency variation that u has.

  Thank you for this comment. In our original manuscript, we were looking at $\overline{u}_{yy}$ at day 0. We have now modified our analysis and take $\overline{u}_{yy}$ averaged over day -10 to -1. We hope the reviewer would agree with us that the 10-day averaged value that is now used in our modified manuscript would help averaging out the high frequency variability contribution.

- Also figure 7, I do not see the LTG-xx lines differ from each other much, what is the purpose of showing them as a multiple panel figure if they can effectively be explained by a sentence or two?

  Thank you for this comment. In Figure 7, we plot the evolution of different variables in different panels, in order compare the evolution of the events identified in the model with the reanalysis events. The lead times are plotted together in the same panel to see if there are any differences in the evolution in the model at different lead times. We have further combined the LTGs following a suggestion by reviewer 1. As we see in Figure 7 that the LTGs lines overlap, this indicates that the mechanisms of the events are well represented in the model for all lead times.

- Figure 8, it is evident that the distributions of uyy have larger variance than those of u. This implies that uyy estimated in this study is not a good measure of waveguide. By definition, a waveguide for stationary Rossby waves should be slow varying. It appears to be measure of PV sharpening on top of background waveguide. Thus, the mechanism that the authors want to study is not captured by uyy.

  Thank you for this comment. In the original manuscript, we use the value of $\overline{u}_{yy}$ at day 0, which is before at the start of the event. As mentioned in earlier replies, we have now modified the $\overline{u}_{yy}$ metric by average day -10 to -1, which we hope to have addressed the problem of high frequency variations. As from the correlation plots shown in Figures 5 to 8 in this document, we find high correlations between $\overline{u}_{yy}$ and $\overline{u}_{zz}$, and between $\overline{u}_{yy}$ and $\overline{q}_y$, respectively. We therefore think for the specific region and level we are looking at, i.e. 55-75°N, 10 hPa, the second term of the meridional PV gradient, $\overline{u}_{yy}$, is a good approximation to the meridional PV gradient and can capture the mechanisms we want to study.

- Figure 9, panels a, b and c of this figure once again suggest that uyy is not a good measure of background waveguide, opposite to what the authors claimed. Its variability is largely associated with wave breaking on both flanks of the polar vortex. Also, this figure be simplified, and the correlations can be summarized by a table or a couple of sentences.

  Thank you for this comment. We think given that $\overline{u}_{yy}$ in the region of investigation has a high correlation with $\overline{q}_y$, $\overline{u}_{yy}$ can be a good measure of the background waveguide. The significant correlation in Figure 9 between the CRPS of $\overline{u}_{yy}$ and $\Delta\overline{u}$ indicates that an improvement in the representation of $\overline{u}_{yy}$ by the model can improve the representation of $\Delta\overline{u}$, though the correlation is not high and thus the influence of $\overline{u}_{yy}$ is not as strong as $\overline{v'T'}$ on predicting $\Delta\overline{u}$. We think the correlation plots can show where each event lies and can point out the two split SSW events which we find might be informative to readers.

[revised manuscript text omitted]

---

## Author Response (AR2)

**Resubmission of wcd-2021-84: Differences in the Sub-seasonal Predictability of Extreme Stratospheric Events**

Dear Co-Editor Yang Zhang,

On behalf of all authors I would like to submit the revised version of the original article "Differences in the Sub-seasonal Predictability of Extreme Stratospheric Events" including an annotated manuscript and a modified version of the manuscript. To address the concerns of referee 2, we have included comments on nonlinear processes in SSW prediction (Lines 446-454) and have included the two suggested references in Line 450.

Once again, thank you very much for your time in reviewing our manuscript.

Best regards,
Rachel Wu

**Reviewer 2**  Specific comments.

1) Line 436: the split SSW event in 2009 has been studied in depth by Gray et al. (2020; 2022). Using controlled modelling simulations, they showed that event involved a series of build-up of acceleration events and nonlinear wave-wave interactions. It is worth citing these more recent papers here. These papers suggest that one of the potential reasons for the limited success in model predicting SSWs is nonlinear processes in the equatorial upper stratosphere during early winter.

Gray, L.J., Brown, M.J., Knight, J. et al. Forecasting extreme stratospheric polar vortex events. Nat Commun 11, 4630 (2020). https://doi.org/10.1038/s41467-020-18299-7.

Gray, L.J., Lu, H., Brown, M.J., Knight, J.R. and Andrews, M.B. (2022), Mechanisms of influence of the Semi-Annual Oscillation on stratospheric sudden warmings. Q J R Meteorol Soc, 148: 1223-1241. https://doi.org/10.1002/qj.4256

Thank you for your comment. We now added the discussion on the nonlinear processes in Lines 447-455 and included the two suggested references.

Lines 446-454: "Furthermore, vortex preconditioning may also be an important factor in determining predictability. For example, certain geometrical configurations of the initial state of the vortex might be more susceptible to vertical wave propagation and weak breaking (Albers and Birner, 2014; Matthewman and Esler, 2011; Esler and Matthewman, 2011). Remote precursors from the tropical stratosphere (Garfinkel et al., 2018; Gray et al., 2020, 2022), the tropical troposphere (Domeisen et al., 2015; Garfinkel and Schwartz, 2017), and the extratropical troposphere (Martius et al.,2009; Karpechko et al., 2018; White et al., 2019; Peings, 2019) can also further enhance predictability on sub-seasonal to seasonal timescales. As such, one might want to investigate how the 2009 and 2018 split SSW events differ from other deceleration events in terms of their preconditioning processes, and to see whether the mechanisms, in particular the nonlinear processes, associated with the two events are well represented in the model."